# SCALABLE SINGLE-CELL GENE EXPRESSION GENERATION WITH LATENT DIFFUSION MODELS

## ABSTRACT

Computational modeling of single-cell gene expression is crucial for understanding cellular processes, but generating realistic expression profiles remains a major challenge. This difficulty arises from the count nature of gene expression data and complex latent dependencies among genes. Existing generative models often impose artificial gene orderings or rely on shallow neural network architectures. We introduce a scalable latent diffusion model for single-cell gene expression data, which we refer to as scLDM, that respects the fundamental exchangeability property of the data. Our VAE uses fixed-size latent variables leveraging a unified Multi-head Cross-Attention Block (MCAB) architecture, which serves dual roles: permutation-invariant pooling in the encoder and permutation-equivariant unpooling in the decoder. We enhance this framework by replacing the Gaussian prior with a latent diffusion model using Diffusion Transformers and linear interpolants, enabling high-quality generation with multi-conditional classifier-free guidance. We show its superior performance in a variety of experiments for both observational and perturbational single-cell data, as well as downstream tasks like cell-level classification.

## 1 INTRODUCTION

Single-cell transcriptomics has revolutionized our understanding of cellular heterogeneity and biological processes at unprecedented resolution (Rozenblatt-Rosen et al., 2017), enabling high-throughput gene expression profiling across millions of cells (Virshup et al., 2023), and providing insights into cellular differentiation (Gulati et al., 2020), disease progression (Zeng & Dai, 2019), responses to drug perturbations (Adduri et al., 2025; Bereket & Karaletsos, 2023; Zhang et al., 2025). However, modeling the complex, high-dimensional gene expression data from single cells presents significant computational and methodological challenges (Lähnemann et al., 2020; Luecken et al., 2022; Neu et al., 2017).

Deep generative modeling (Tomczak, 2024) offers a powerful framework to formulate expressive probability distributions. In the context of single-cell data, multiple methods have been proposed. In particular, Variational Auto-Encoders (VAEs) have been extensively utilized for representation learning (single-cell Variational Inference; scVI) (Lopez et al., 2018), perturbation modeling (Lotfollahi et al., 2023b; Palma et al., 2025b), trajectory inference (Gayoso et al., 2024), among others (Gayoso et al., 2022). Additionally, Generative Adversarial Networks (GANs) have also been proposed, both for generating realistic cell populations (scGAN; (Marouf et al., 2020b)) and for inferring cellular trajectories (Reiman et al., 2021). Recently, diffusion-based models have also been adopted for single-cell gene expression (Luo et al., 2024). An interesting research line was proposed in (Palma et al., 2025a) that combines scVI with a flow matching model in the latent space (CFGen).

However, two key challenges limit existing methods. First, they often require a fixed ordering of genes or operate on a restricted subset of highly variable genes (HGVs). This assumption directly clashes with the biological reality that gene expression profiles are **exchangeable** sets, where the order of genes carries no meaning. Second, approaches based on GANs inherit well-known training instabilities and risks of mode collapse. These limitations make current models inflexible, difficult to scale, and unable to properly handle the unordered nature of single-cell data.

This paper introduces a novel approach that combines the flexibility of VAEs with the power of latent diffusion models (see Figure 1), specifically designed to handle the exchangeable nature of gene

expression data. The key insight is that careful architectural choices, particularly in the parameterization of permutation-invariant and permutation-equivariant components, result in a scalable, deep, and exchangeable generative model. The contributions of the paper are the following:

- We propose a novel fully transformer-based VAE architecture for exchangeable data that uses a single set of fixed-size, permutation-invariant latent variables. The model utilizes a Multi-head Cross-Attention Block (MCAB) that serves two purposes: It acts as a permutation-invariant pooling operator in the encoder, and functions as a permutation-equivariant unpooling operator in the decoder. This unified approach eliminates the need for separate architectural components for handling varying set sizes.

- We replace the standard Gaussian prior with a latent diffusion model trained with the flow matching loss and linear interpolants using the Scalable Interpolant Transformers formulation (SiT) (Ma et al., 2024), and a denoiser parameterized by Diffusion Transformers (DiT) (Peebles & Xie, 2023). This allows for better modeling of the complex distribution of cellular states and enables controlled generation through classifier-free guidance.

- The proposed framework, which we refer to as scLDM, supports generation conditioned on multiple attributes simultaneously through an extended classifier-free guidance mechanism, enabling fine-grained control over generated cell states, as demonstrated on multiple benchmark datasets. Moreover, we indicate the strengths of our fully transformer-based auto-encoder in terms of reconstruction metrics and on a downstream prediction task.

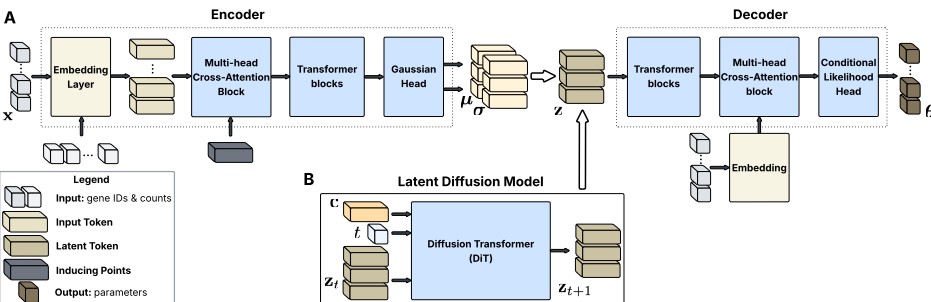

Figure 1: Our deep generative model, scLDM, for single-cell gene expression data. **A**: A fully transformer-based architecture for processing gene expressions. The encoder network results in permutation-invariant latent variables represented as tokens. The decoder network returns permutation-equivariant counts for given gene IDs. **B**: At the second stage, a vanilla prior is replaced by a latent diffusion model. We model latent tokens using Diffusion Transformers (DiT), and train the resulting LDM using linear interpolants and the flow matching loss. Sampling is carried out by applying the Scalable Interpolant Transformers (SiT) library (Ma et al., 2024).

## 2 BACKGROUND

**Variational Auto-Encoders** Another approach is Variational Auto-Encoders (Kingma & Welling, 2014; Rezende et al., 2014), which offer flexible modeling capabilities. (Kim et al., 2021) proposed SetVAE with two latent variables for varying set sizes: $\mathbf{z}_\mathcal{I}$ matching $\mathbf{x}_\mathcal{I}$'s dimensionality (where $\mathbf{z}_i$ corresponds to $\mathbf{x}_i$, $i \in \mathcal{I}$) and constant-size $\mathbf{c} \in \mathbb{R}^{d_1}$. They used hierarchical VAE with multiple $\mathbf{z}_\mathcal{I}$ and $\mathbf{c}$ layers and replaced conditional likelihood with Chamfer Distance. While we appreciate VAE's flexibility, we find two distinct latents and hierarchical structure unnecessary, arguing that careful parameterization is *crucial* for high performance.

**Permutation-equivariant/invariant Parameterizations** Geometric deep neural networks typically compose permutation-invariant and/or permutation-equivariant layers with nonlinearity activations (Bronstein et al., 2021). DeepSets (Zaheer et al., 2017) exemplifies this blueprint by processing elements consistently regardless of position, then applying symmetric aggregation (averaging or pooling (Kimura et al., 2024; Ilse et al., 2018; Xie & Tong, 2025)) to ensure permutation invariance. However, processing elements separately before aggregation with non-learnable pooling is limiting. Learned attention mechanisms in transformer architectures offer a solution, enabling joint element

transformation. SetTransformer (Lee et al., 2019) introduces multi-head attention blocks and Pooling by Multi-head Attention for permutation invariance. We propose an alternative parameterization using a single multi-head attention layer for fixed-size output, followed by transformer blocks.

**Latent Diffusion Models** Latent Diffusion Models (LDMs) perform diffusion processes in learned latent spaces rather than directly in high-dimensional data spaces. Stable Diffusion (Rombach et al., 2022) pioneered this approach for text-to-image synthesis by training diffusion models in the latent space of a pre-trained VAE, dramatically reducing computational costs while maintaining generation quality. This paradigm has proven effective across diverse scientific domains: all-atom diffusion transformers (Joshi et al., 2025) generate molecules and materials with atomic-level precision, similarly LaM-SLidE (Sestak et al., 2025) utilizes transformer-based LDM for molecular dynamics (among others), while La-proteina (Geffner et al., 2025) employs transformer-based partially latent flow matching for atomistic protein generation. These advances demonstrate the versatility of latent diffusion approaches for complex, high-dimensional scientific data across multiple modalities. Here, we extend this framework to single-cell transcriptomics by proposing a transformer-based LDM for this biological data type.

**Generative Models for scRNA-seq** In the context of single-cell genomics, numerous generative models have been developed for (conditional) sampling of gene expression profiles. scVI (Lopez et al., 2018) represents an early VAE-based generative model, while more recent approaches include GAN-based and diffusion-based architectures such as scGAN (Marouf et al., 2020a) and scDiffusion (Luo et al., 2024). These models operate in continuous space and therefore transform discrete gene expression data into log-normalized counts. Recently, latent diffusion frameworks have emerged with models like SCLD (Wang et al., 2023) and CFGen (Palma et al., 2025a), which leverage latent diffusion frameworks. Additionally, application-specific generative models have been developed for perturbational single-cell genomics, including CPA Lotfollahi et al. (2023a), SquiDiff (He et al., 2024), CellFlow (Klein et al., 2025), and CellOT (Bunne et al., 2023), which are tailored to capture the effects of genetic and chemical perturbations on cellular states. Our approach is similar in vein to CFGen and SCLD, but leverages transformer-based architectures for both our newly proposed VAE as well as the latent diffusion model.

# 3 METHODOLOGY

## 3.1 PROBLEM FORMULATION

Let us consider $M$ random variables, $\mathbf{x}$, where each $\mathbf{x}_i \in \mathbb{X}^D$, e.g., $\mathbb{X} = \mathbb{N}$. A set of indices of $M$ random variables is denoted as $\mathcal{I}$, namely, $\mathcal{I} = \pi(\{1, 2, \ldots, M\})$, where $\pi(\cdot)$ is a permutation[1]. Further, we denote a specific order of variables in $\mathbf{x}$ determined by $\mathcal{I}$ as $\mathbf{x}_\mathcal{I}$. We assume that for a given $\mathcal{I}$, an object $\mathbf{x}_\mathcal{I}$ is equivalent to an object defined by $\pi(\mathcal{I})$, namely, $\mathbf{x}_\mathcal{I} = \mathbf{x}_{\pi(\mathcal{I})}$. An example of such a setting is gene expression data where $\{1, 2, \ldots, M\}$ corresponds to gene IDs and the order of gene IDs does not change the state of a cell. Further, we assume a *true* conditional distribution model $p(\mathbf{x}_\mathcal{I}|\mathcal{I})$ that for a given order of indices $\mathcal{I}$ allows sampling $\mathbf{x}_\mathcal{I}$. We access this *true* distribution through observed *iid* data $\mathcal{D} = \{(\mathbf{x}_{\mathcal{I}_n}, \mathcal{I}_n)\}_{n=1}^N$. We look for a model $p(\mathbf{x}_\mathcal{I}|\theta, \mathcal{I})$ with parameters $\theta$ that optimizes the log-likelihood function for the empirical distribution with data $\mathcal{D}$, $\ell(\theta; \mathcal{D}) = \sum_{n=1}^N \ln p(\mathbf{x}_{\mathcal{I}_n}|\theta, \mathcal{I}_n)$. Moreover, we are interested in finding a single model that for given indices $\mathcal{I}$ generates corresponding $\mathbf{x}_\mathcal{I}$. Formally, we require the model to be *exchangeable*, namely, $p(\mathbf{x}_\mathcal{I}|\mathcal{I}) = p(\mathbf{x}_{\pi(\mathcal{I})}|\pi(\mathcal{I}))$. For instance, a model generates the same gene expression profile for given different orders of gene IDs.

To model an exchangeable probabilistic model $p(\mathbf{x}_\mathcal{I}|\theta, \mathcal{I})$, we introduce $m$ latent variables (i.e., the number of latents is fixed for all subsets $\mathcal{I}$), $\mathbf{Z} \in \mathbb{R}^{m \times D}$. By using the family of variational posteriors of the form $q(\mathbf{Z}|\phi, \mathbf{x}_\mathcal{I})$, the Evidence Lower BOund (ELBO) is the following:

$$\ln p(\mathbf{x}_\mathcal{I}|\theta, \mathcal{I}) \geq \mathbb{E}_{\mathbf{Z} \sim q(\mathbf{Z}|\phi, \mathbf{x}_\mathcal{I})} [\ln p(\mathbf{x}_\mathcal{I}|\eta, \mathbf{Z}, \mathcal{I}) + \ln p(\mathbf{Z}|\psi) - \ln q(\mathbf{Z}|\phi, \mathbf{x}_\mathcal{I})], \quad (1)$$

where $\theta = \{\eta, \psi, \phi\}$ are the parameters of the model. We propose to model these parameters using neural networks, namely: $\phi(\mathbf{x}_\mathcal{I}) = \text{NN}_{enc}(\mathbf{x}_\mathcal{I})$, $\eta(\mathbf{Z}, \mathcal{I}) = \text{NN}_{dec}(\mathbf{Z}, \mathcal{I})$, and $\psi$ are weights of a parameterization of the prior. Since our assumption is that the model must be exchangeable, we

---

[1]We denote a permutation either as a function $\pi(\cdot)$ or, equivalently, as a matrix $\mathbf{P}$.

propose to parameterize the distributions in a way that: (i) $\mathbf{Z}$ is permutation-invariant, namely, we aim for defining variational posteriors as Gaussian distributions with permutation-invariant neural networks $\mathrm{NN}_{enc}$, (ii) the conditional likelihood is defined as $p(\mathbf{x}_{\mathcal{I}}|\eta(\mathbf{Z},\mathcal{I})) = \prod_{i \in I} p(\mathbf{x}_i|\eta_i(\mathbf{Z},\mathcal{I}))$, hence, we must ensure that: $\mathbf{P}\eta(\mathbf{Z}, \pi(\mathcal{I})) = \mathrm{NN}(\mathbf{Z}, \pi(\mathcal{I}))$.

## 3.2 SCLDM: A TRANSFORMER-BASED VAE WITH LATENT DIFFUSION

**Permutation-invariant/equivariant Cross-Attention** Our VAE is parameterized by a fully transformer-based architecture that leverages multi-head cross-attention block (MCAB), enabling pooling/unpooling operations to avoid processing tens of thousands of tokens at the same time:

$$\mathrm{MCAB}_{\mathbf{S}}(\mathbf{X}) = F(\mathbf{X}, \mathbf{S}) + \mathrm{MLP}(\mathrm{LN}_F(F(\mathbf{X}, \mathbf{S}))) \tag{2}$$

$$F(\mathbf{X}, \mathbf{S}) = \mathbf{Q} + \mathrm{Att}_K(\mathrm{LN}_Q(\mathbf{Q}), \mathbf{K}, \mathbf{V})) \tag{3}$$

$$\mathbf{Q} = \mathrm{Linear}_S(\mathbf{S}), \ \mathbf{K} = \mathrm{Linear}_K(\mathrm{LN}_K(\mathbf{X})), \ \mathbf{V} = \mathrm{Linear}_V(\mathrm{LN}_V(\mathbf{X})), \tag{4}$$

where Linear is a linear layer, $\mathrm{LN}(\cdot)$ denotes a layer norm, and $\mathrm{MLP}(\cdot)$ is a small fully-connected neural network, e.g., $\mathrm{MLP}(\mathbf{X}) = (\mathrm{Linear} \circ (\mathrm{Linear} \odot (\mathrm{silu} \circ \mathrm{Linear}))(\mathbf{X}).$[2] $\mathbf{S}$ are learnable pseudoinputs. $\mathrm{MCAB}_{\mathbf{S}}$ is defined similarly to a block used in Perceiver (Jaegle et al., 2022; 2021).

MCAB is either permutation-invariant or permutation-equivariant. Since it relies on the attention mechanism, if we permute $\mathbf{X}$ but do not permute $\mathbf{S}$, then MCAB is permutation-invariant (see Property 3 for the proof). However, if we process $\mathbf{Z}$ by a permutation-invariant function and we permute $\mathbf{S}$ accordingly to the permuted indices, then MCAB becomes permutation-equivariant (see Property 4 for the proof). As a result, we use MCAB as a permutation-invariant pooling operator in the encoder network, and as a permutation-equivariant unpooling operator in the decoder network.

**Encoder (Variational Posterior)** We define the family of variational posteriors as Gaussians, $q(\mathbf{Z}|\phi(\mathbf{x}_{\mathcal{I}})) = \mathcal{N}(\mathbf{Z}|\mu(\mathbf{x}_{\mathcal{I}}), \sigma(\mathbf{x}_{\mathcal{I}})), \phi(\mathbf{x}_{\mathcal{I}}) \stackrel{df}{=} \{\mu(\mathbf{x}_{\mathcal{I}}), \sigma^2(\mathbf{x}_{\mathcal{I}})\}$. We need $\mathbf{Z}$ to be of fixed size and invariant to permutations of $\mathbf{x}_{\mathcal{I}}$, we propose the following architecture of the encoder network:

$$\mathrm{NN}_{enc}(\mathbf{x}_{\mathcal{I}}, \mathcal{I}) = (\mathrm{T}_L \circ \mathrm{T}_{L-1} \circ \ldots \circ \mathrm{T}_1 \circ \mathrm{MSCAB}_{\mathbf{S}} \circ \mathrm{Embedding})(\mathbf{x}_{\mathcal{I}}, \mathcal{I}), \tag{5}$$

where $\mathrm{T}_l(\cdot)$ denotes a transformer block, e.g., $\mathrm{T}_l(\mathbf{X}) = ((\mathrm{Id} \oplus (\mathrm{MLP} \circ \mathrm{LN}_2)) \circ (\mathrm{Id} \oplus (\mathrm{Att}_K \circ \mathrm{LN}_1)))(\mathbf{X})$, and $\mathrm{Embedding}(\cdot, \cdot)$ is an embedding layer. Since inputs $\mathbf{x}_{\mathcal{I}}$ form a (column) vector of counts, and $\mathcal{I}$ are IDs, we propose to use the following embedding layer:

$$\mathrm{Embedding}(\mathbf{x}_{\mathcal{I}}, \mathcal{I}) = \mathrm{Linear} \circ (\mathrm{repeat}_D(\mathbf{x}_{\mathcal{I}}) \boxplus \mathbf{E}_{\mathcal{I}}), \tag{6}$$

where $\mathrm{repeat}_D$ repeats the counts $D$-times resulting in a matrix $M \times D$, Linear projects the concatenated $2D$-dimensional space to the $D$-dimensional space, and $\mathbf{E} \in \mathbb{R}^{M \times D}$ is the embedding matrix. The rationale behind this way of embedding both counts and indices is to mix the information and be able to learn the mixing through a projection layer. Additionally, we propose to encode only expressed genes, and replace non-expressed genes with a PAD token. We provide more details and an example in Appendix E.1.

The last transformer block duplicates the embedding dimension such that both the means $\mu$ and the variances $\sigma^2$ of a Gaussian are modeled. Alternatively, we can output means only to have an autoencoder architecture, which is typically used in Latent Diffusion Models (Rombach et al., 2022). Note that all transformer blocks are permutation-equivariant, but our $\mathrm{MCAB}_{\mathbf{S}}$ is permutation-invariant. As a result, the proposed parameterization $\mathrm{NN}_{enc}$ results in permutation-invariant variational posteriors.

**Decoder (Conditional Likelihood)** The decoder network parameterizes the conditional likelihood function $p(\mathbf{x}_{\mathcal{I}}|\eta(\mathbf{Z},\mathcal{I}))$ for given latents $\mathbf{Z}$ and indices $\mathcal{I}$. The conditional likelihood could be a Gaussian if $\mathbf{x}$'s are continuous, or Poisson or Negative Binomial for counts. To fulfill the requirement on modeling exchangeable distributions, we need to ensure the conditional likelihood is exchangeable. In other words, for a given permutation $\pi$, the following holds true: $p(\mathbf{x}_{\mathcal{I}}|\eta(\mathbf{Z},\mathcal{I})) = p(\mathbf{x}_{\pi(\mathcal{I})}|\eta(\mathbf{Z}, \pi(\mathcal{I})))$. First, we assume that for given $\mathbf{Z}$, the conditional likelihood is fully factorized: $p(\mathbf{x}_{\mathcal{I}}|\eta(\mathbf{Z},\mathcal{I})) = \prod_{i \in I} p(\mathbf{x}_i|\eta_i(\mathbf{Z},\mathcal{I}))$. Next, we make the parameterization of

---

[2] We use the following notation for function compositions: $(f \circ g)(x) \stackrel{df}{=} f(g(x))$, $(f \cdot g)(x) \stackrel{df}{=} f(x)g(x)$, $(f \oplus g)(x) \stackrel{df}{=} f(x) + g(x)$, and $(f \boxplus g)(x) \stackrel{df}{=} \mathrm{concatenate}(f(x), g(x))$.

$p(\mathbf{x}_{\mathcal{I}}|\eta(\mathbf{Z},\mathcal{I}))$ permutation equivariant, because, otherwise, transforming $\mathbf{Z}$ would result in incorrect parameters for each component $p(\mathbf{x}_i|\eta_i(\mathbf{Z},\mathcal{I}))$. Keeping in mind that $\mathbf{Z}$ is permutation-invariant to permutations of $\mathbf{x}_{\mathcal{I}}$, we propose the following decoder network:

$$\mathrm{NN}_{dec}(\mathbf{Z},\mathcal{I}) = (\mathrm{MCAB}_{\mathbf{E}_{\mathcal{I}}} \circ \mathrm{T}_L \circ \ldots \circ \mathrm{T}_1)(\mathbf{Z},\mathcal{I}), \tag{7}$$

and then use the outcomes of $\mathrm{NN}_{dec}(\mathbf{Z},\mathcal{I})$ to parameterize an appropriate distribution, e.g., the Negative Binomial (see Appendix E.2 for further details).

In our decoder network, we use $\mathrm{MCAB}_{\mathbf{E}_{\mathcal{I}}}$ as our final block that outputs the parameters of the conditional likelihood. To make sure the model is permutation-equivariant, we define pseudoinputs in the multi-head cross-attention block selecting embedding vectors specified by $\mathcal{I}$, $\mathbf{S} = \mathbf{E}_{\mathcal{I}}$, where $\mathbf{E}$ is the embedding used in the encoder network. This way, we ensure permutation-equivariance since permuting indices is equivalent to permuting embedding vectors, $\mathbf{E}_{\pi(\mathcal{I})} = \mathbf{E}_{\mathcal{I}}$, see Property 4 in Appendix. Eventually, we obtain a family of exchangeable conditional likelihood functions.

**Prior (Marginal over Latents)** The final component of the proposed VAE is the *prior* of latent variables. Formulating permutation-equivariant priors is challenging (Kuzina et al., 2022); fortunately, our latents $\mathbf{Z}$ are permutation-invariant and length-invariant. As a result, we can use any prior distribution we prefer, including standard Gaussian, $p(\mathbf{Z}) = \mathcal{N}(\mathbf{Z}|\mathbf{0},\mathbf{I})$.

In this paper, we advocate to use a Latent Diffusion Model (LDM) (Rombach et al., 2022), namely, for a pre-trained VAE, we fit a diffusion-based model in the latent space to replace a *simpler* prior like $\mathcal{N}(\mathbf{Z}|\mathbf{0},\mathbf{I})$. Using LDMs not only results in a better match with the aggregated posterior (Tomczak & Welling, 2018; Tomczak, 2024), but allows the application of controlled sampling using techniques such as classifier-free guidance (Ho & Salimans, 2022). In particular, we focus on linear interpolants and the flow matching (FM) loss Lipman et al. (2022); Tong et al. (2023), and the following version of the classifier-free guidance for FM:

$$\tilde{v}_{t,\epsilon}(\mathbf{Z},y) = v_{t,\epsilon}(\mathbf{Z};\mathrm{Null}) + \omega\left[v_{t,\epsilon}(\mathbf{Z};y) - v_{t,\epsilon}(\mathbf{Z};\mathrm{Null})\right], \tag{8}$$

where $v_{t,\epsilon}(\mathbf{Z};\cdot)$ is a parameterized vector field, and $\omega$ is the guidance strenght for attributes $\mathbf{y} \in \{0,1\}^J$, where any combination of attributes is possible (we refer to it as *joint conditioning*); the Null attribute corresponds to no conditioning. In CFGen (Palma et al., 2025a), a different classifier-free guidance was used, namely, $\tilde{v}_{t,\epsilon}(\mathbf{Z},y) = v_{t,\epsilon}(\mathbf{Z};\mathrm{Null}) + \sum_{j=1}^{J}\omega_j\left[v_{t,\epsilon}(\mathbf{Z};y_j) - v_{t,\epsilon}(\mathbf{Z};\mathrm{Null})\right]$, that assumes *additive conditioning* s.t. $\sum_j y_j = 1$.

We parameterize the vector field (score) model using Diffusion Transformer (DiT) blocks (Peebles & Xie, 2023). The network is a composition of DiT and perfectly fits our modeling scenario since latents $\mathbf{Z}$ are tokens.

## 3.3 Training & Sampling

**Training** We train our model (scLDM) using the two-stage approach: (1) A VAE is trained to learn a permutation-invariant latent space by reconstructing subsets of variables; and (2) An LDM is trained to generate new samples from this latent space which can be controlled by classifier-free guidance (Ho & Salimans, 2022) with multiple conditions (Palma et al., 2025a).

*Stage 1: VAE* We train our VAE with a standard Gaussian prior by optimizing the ELBO in equation 1. However, to encourage better reconstruction capabilities, we introduce $\beta$-weighting of the KL-term like in (Higgins et al., 2017). In the most extreme case, we set $\beta$ to 0 and the encoder returns means only, $\mu(\mathbf{x}_{\mathcal{I}})$.

*Stage 2: LDM* In the second stage, we freeze the VAE and replace the standard Gaussian prior with a score-based (diffusion) model parameterized by a DiT network trained with linear interpolants and the flow matching loss. Additionally, to encourage controlled sampling, for each element of a mini-batch, we sample from the Bernoulli distribution with probability $\rho$ to determine whether conditioning is used or not.

**Sampling** In our model, sampling $\mathbf{x}$'s determined by the indices $\mathcal{I}$ is defined by the following generative process: (i) $\mathbf{Z} \sim p(\mathbf{Z})$, (ii) $\mathbf{x}_{\mathcal{I}} \sim p(\mathbf{x}_{\mathcal{I}}|\eta(\mathbf{Z},\mathcal{I}))$. We can also sample *conditionally* by applying the classifier-free guided sampling technique, following the vector field defined in equation 8.

# 4 EXPERIMENTS

**Settings** We provide more details on the experiments in the Appendix, namely, the datasets in Appendix F, the baselines in Appendix G, the hyperparams of our scLDM in Appendix H, the evaluation pipeline with metrics in Appendix I, and additional results in Appendix K. In the following experiments, we present superior capabilities of our scLDM: (i) the powerful reconstructive performance of the fully transformer-based VAE, (ii) the unconditional and conditional generative performance on observational and perturbational datasets, (iii) the usefulness of the embeddings provided by our auto-encoder on classification downstream tasks.

## 4.1 (UN)CONDITIONAL CELL GENERATION ON OBSERVATIONAL DATA

**Details** For the first experiment, we used single-cell RNA-sequencing data from the benchmark datasets used in (Palma et al., 2025a). Here, we are interested in evaluating the reconstructive and generative capabilities of our scLDM. For generations, we train our scLDM to synthesize gene expression profiles conditioned on a single attribute. At inference time, we query the model with specific labels to generate new synthetic cells that match the desired cellular identity. In the case of unconditional generation, we sample from the vector field without conditioning on the cell type label (i.e., $y = \text{Null}$). We compare our approach to scVI (Lopez et al., 2018), scDiffusion (Luo et al., 2024), and the current SOTA generative model CFGen (Palma et al., 2025a).

**Results and discussion** Our proposed scLDM model demonstrates substantial improvements over existing approaches across all evaluated datasets and metrics, see Table 1. scLDM consistently achieves the lowest reconstruction error values, with particularly notable improvements on Tabula Muris (4569.6 vs. 5547.6 for CFGen) and HLCA (4102.1 vs. 5428.7 for CFGen) datasets. The Pearson correlation coefficients show dramatic improvements, with scLDM achieving 0.391 on Tabula Muris com-

Table 1: Model performance comparison on cell reconstruction task.

| Dataset | Model | RE ↓ | PCC ↑ | MSE ↓ |
|---|---|---|---|---|
| Dentate Gyrus | scVI | **5193.2** ± 0.1 | 0.058 ± 0.000 | 0.378 ± 0.000 |
| | CFGen | 5468.8 ± N/A | 0.076 ± N/A | 0.253 ± N/A |
| | scLDM | **5232.9** ± 43.1 | **0.103** ± 0.005 | **0.249** ± 0.002 |
| Tabula Muris | scVI | 5588.2 ± 1.7 | 0.221 ± 0.000 | 0.132 ± 0.000 |
| | CFGen | 5547.6 ± N/A | 0.136 ± N/A | 0.127 ± N/A |
| | scLDM | **4569.6** ± 105.1 | **0.391** ± 0.021 | **0.092** ± 0.004 |
| HLCA | scVI | 5659.2 ± 0.5 | 0.125 ± 0.000 | 0.238 ± 0.000 |
| | CFGen | 5428.7 ± N/A | 0.146 ± N/A | 0.117 ± N/A |
| | scLDM | **4102.1** ± 41.1 | **0.421** ± 0.013 | **0.069** ± 0.001 |

pared to 0.221 for scVI and 0.136 for CFGen, nearly doubling the correlation with ground truth. Similarly, MSE is consistently reduced, with scLDM achieving 0.069 on HLCA compared to 0.117 for CFGen and 0.238 for scVI. These results suggest that our fully transformer-based VAE is able to more effectively capture the complex structure of single-cell gene expression data compared to traditional VAE-based methods (scVI, CFGen). The consistent improvements across diverse tissue types (brain, entire organism, and lung) indicate the generalizability of our approach, namely, a parameterization of the VAE using the proposed transformer-based architectures.

Table 2 presents the generation benchmarks, where scLDM demonstrates superior performance across both unconditional and conditional generation sampling. In the unconditional setting, our model achieves the lowest Wasserstein-2 distance across all datasets, with improvements ranging from 14% on Dentate Gyrus to 12% on Tabula Muris. While CFGen shows competitive performance on $\text{MMD}^2$ RBF, our approach matches or outperforms it, achieving identical scores on HLCA and superior results on Tabula Muris. In terms of the Fréchet Distance (FD), scLDM still shows superior performance, with particularly striking improvements on Tabula Muris, where it achieves a nearly three-fold reduction compared to the second-best baseline. For conditional generation, scLDM maintains its perfor-

Table 2: Model performance comparison on (un)conditional cell generation benchmarks on highly variable genes.

| Setting | Model | W2 ↓ | $\text{MMD}^2$ RBF ↓ | FD ↓ |
|---|---|---|---|---|
| | | Dentate Gyrus | | |
| Uncond | scDiffusion | 17.443 ± 0.028 | 0.258 ± 0.002 | 256.630 ± 0.357 |
| | CFGen | 12.617 ± 0.034 | **0.022** ± 0.001 | **28.105** ± 0.332 |
| | scLDM | **10.817** ± 0.065 | 0.023 ± 0.000 | 28.403 ± 0.099 |
| Cond | scDiffusion | 17.321 ± 0.041 | 0.689 ± 0.000 | 261.217 ± 1.856 |
| | CFGen | 11.608 ± 0.066 | **0.075** ± 0.000 | 41.425 ± 1.612 |
| | scLDM | **10.615** ± 0.028 | 0.102 ± 0.003 | **34.388** ± 1.014 |
| | | Tabula Muris | | |
| Uncond | scDiffusion | 14.143 ± 0.007 | 0.144 ± 0.001 | 158.977 ± 1.070 |
| | CFGen | 11.658 ± 0.127 | 0.008 ± 0.000 | 36.373 ± 1.165 |
| | scLDM | **10.295** ± 0.110 | **0.004** ± 0.000 | **13.130** ± 0.318 |
| Cond | scDiffusion | 14.143 ± 0.007 | 0.144 ± 0.001 | 158.977 ± 1.070 |
| | CFGen | 8.921 ± 0.034 | 0.026 ± 0.000 | 21.517 ± 0.596 |
| | scLDM | **7.717** ± 0.030 | **0.016** ± 0.000 | **11.008** ± 0.716 |
| | | HLCA | | |
| Uncond | scDiffusion | 15.886 ± 0.038 | 0.163 ± 0.001 | 210.853 ± 1.165 |
| | CFGen | 12.433 ± 0.045 | **0.007** ± 0.000 | 24.639 ± 0.738 |
| | scLDM | **10.419** ± 0.079 | **0.007** ± 0.000 | **18.024** ± 0.372 |
| Cond | scDiffusion | 15.886 ± 0.038 | 0.163 ± 0.001 | 210.853 ± 1.165 |
| | CFGen | 9.757 ± 0.078 | 0.090 ± 0.006 | 33.900 ± 5.116 |
| | scLDM | **8.445** ± 0.045 | **0.074** ± 0.002 | **20.974** ± 1.504 |

mance edge with consistent improvements in W2, $\text{MMD}^2$ RBF, and FD scores across all datasets.

We report further generation results on all genes in Appendix. In Figure 2 we report qualitative evaluations of generation results for the HLCA datasets for all three models. Our model shows qualitatively a better coverage of the cell state variation on UMAP coordinates, showcasing how it is able to recapitulate high resolution cell states in highly heterogenous tissues like the human lung. Additionally, in Appendinx K.2 we provide an interpretability analysis on the cross-attention scores of the encoder-decoder model of scLDM, showing how the latent tokens map to specific marker gene set patterns. These results demonstrate that our latent diffusion approach not only generates more realistic single-cell expression profiles but also maintains superior performance when conditioning on cell state information, a crucial capability for practical applications in single-cell genomics.

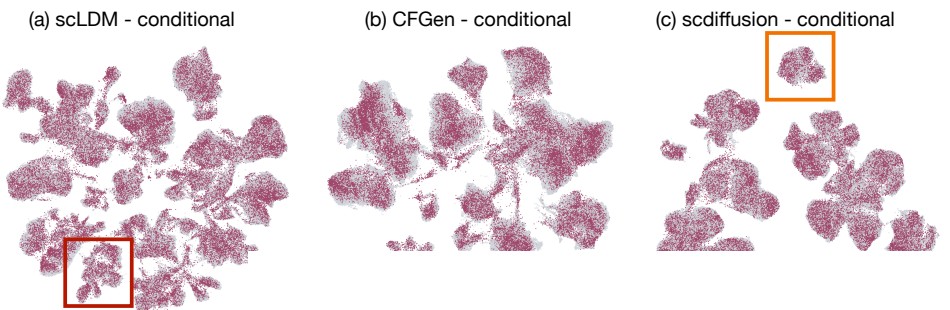

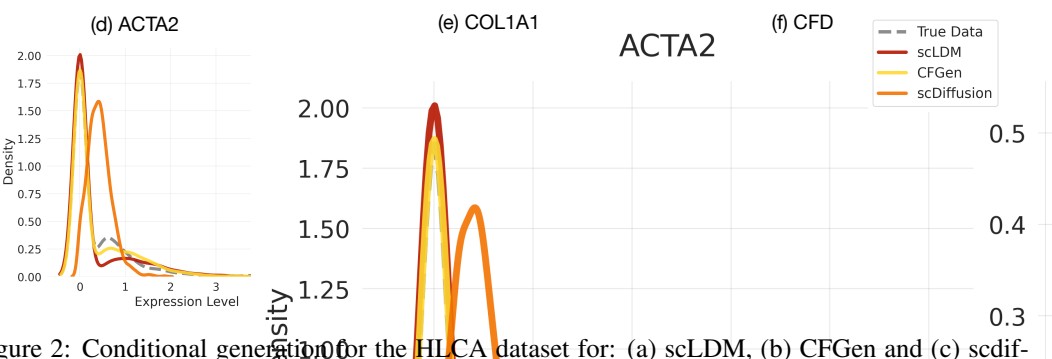

Figure 2: Conditional generation for the HLCA dataset for: (a) scLDM, (b) CFGen and (c) scdiffusion. Expression levels for 3 marker genes: (d) ACTA2, (e) COL1A1 and (f) CFD, markers of "alveolar type 2 fibroblast cell", corresponding to cell populations in the insets.

## 4.2 CONDITIONAL CELL GENERATION ON PERTURBATIONAL DATA

**Details** In the second experiment, we train our model for conditional gene expression generation based on multiple attributes: a cell context (cell lines and cell types) and a perturbation type (gene knockouts and cytokines). The VAE baseline is trained without attribute conditioning, focusing solely on the reconstruction objective, while the flow matching component incorporates multi-attribute conditioning. By training across diverse contexts, the model learns to capture joint structure spanning different axes of variation. At inference time, the flow matching model is queried with specific combinations of cell type and perturbation to generate new gene expression profiles.

We leverage two datasets: (1) Parse 1M, containing perturbational single-cell RNA-sequencing data from human peripheral blood mononuclear cells (PBMCs) generated by Parse Biosciences (par) with 1,267,690 single cells across 18 annotated cell types, each subjected to one of 90 cytokine perturbations or a control condition, and to test generalization capabilities, we hold out 27 cytokine perturbations in CD4 Naive cells; and (2) Replogle, a benchmark genetic perturbation dataset (Nadig et al., 2025) consisting of 2,024 gene knockouts across four cell lines after filtering perturbations with low on-target efficacy Adduri et al. (2025), holding out 372 genetic perturbations in HepG2 cells to evaluate generalization to unseen cell context–perturbation pairs. For both datasets, we restricted analysis to the top 2,000 highly variable genes (HVGs) following Adduri et al. (2025). We compare our model against established baselines: CPA (Lotfollahi et al., 2023a), scVI (Lopez et al., 2018), scGPT (Cui et al., 2024), STATE-Tx (Adduri et al., 2025) and CellFlow (Klein et al., 2025).

**Results and Discussion** The results presented in Table 3 demonstrate that our proposed approach significantly outperforms the baselines in both the Parse 1M dataset (cytokine perturbation) and the Replogle dataset (gene knockouts). Our model scLDM is substantially better across all metrics, improving up to ~90% for MMD$^2$ RBF and FD for the Parse 1M dataset and ~60% for MMD$^2$ RBF and FD for the Replogle dataset. This demonstrates how our model outperforms others in capturing the full range of cellular variation in perturbation responses across unseen combinations of cell contexts and perturbations. In Appendix 14, we report four additional metrics on perturbation

Table 3: Model performance comparison on conditional cell generation on Parse1M and Replogle. For scLDM, we evaluated the generation performance across 3 different guidance weights ($\omega$)

| Dataset | Model | W2 ↓ | MMD$^2$ RBF ↓ | FD ↓ |
|---|---|---|---|---|
| Parse 1M | scVI | 35.508 ± 0.182 | 1.372 ± 0.016 | 1233.109 ± 12.694 |
| | CPA | 13.534 ± 0.036 | 1.117 ± 0.014 | 181.324 ± 0.985 |
| | Cellflow | **11.836** ± 0.063 | **0.015** ± 0.002 | **9.443** ± 1.238 |
| | scGPT | 22.870 ± 0.152 | 2.203 ± 0.013 | 523.932 ± 7.043 |
| | STATE | 19.111 ± 0.136 | 0.714 ± 0.009 | 312.344 ± 5.743 |
| | scLDM ($\omega$=1) | 12.457 ± 0.045 | 0.027 ± 0.002 | 18.136 ± 0.903 |
| | scLDM ($\omega$=5) | 12.902 ± 0.087 | 0.071 ± 0.004 | 43.363 ± 2.246 |
| | scLDM ($\omega$=10) | 13.638 ± 0.111 | 0.122 ± 0.006 | 69.769 ± 3.363 |
| Replogle | scVI | 17.359 ± 0.051 | 0.453 ± 0.003 | 284.474 ± 1.825 |
| | CPA | 11.510 ± 0.029 | 0.532 ± 0.003 | 126.805 ± 0.693 |
| | Cellflow | **10.684** ± 0.046 | 0.289 ± 0.003 | 73.358 ± 0.977 |
| | scGPT | 34.166 ± 0.272 | 3.087 ± 0.010 | 1247.679 ± 20.245 |
| | STATE | 20.582 ± 0.039 | 0.730 ± 0.003 | 366.642 ± 1.547 |
| | scLDM ($\omega$=1) | 11.292 ± 0.033 | **0.200** ± 0.002 | **53.750** ± 0.666 |
| | scLDM ($\omega$=5) | 12.900 ± 0.069 | 0.320 ± 0.004 | 105.365 ± 1.935 |
| | scLDM ($\omega$=10) | 14.911 ± 0.091 | 0.436 ± 0.005 | 166.877 ± 3.036 |

predictions in unseen context, using the `cell-eval` Adduri et al. (2025). Our model is competitive, and sometimes outperforms the stronger baseline STATE-Tx across both datasets.

In Figure 3, we report a qualitative evaluation of our model generative performances for the Parse 1M dataset for unseen combinations of CD4-Naive cells with various cytokine perturbations such as IL-9 and LT-alpha1-beta2. Furthermore, we show the same for Replogle dataset for unseen combinations of HepG2 cells with PPP6c and ZDHHC7 gene edits.

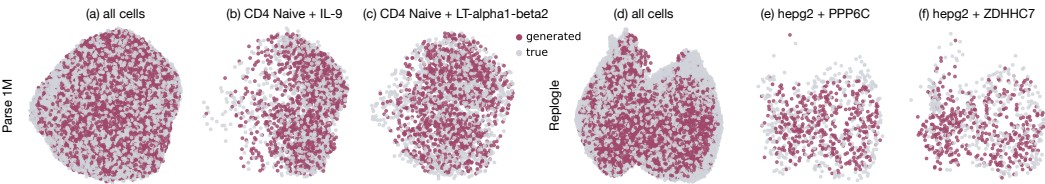

Figure 3: Conditional generation across multiple attributes: cell type and perturbation. (a) Generated vs. true cells across all cell types in the Parse 1M dataset show close alignment. (b–c) For CD4 Naive cells, conditioning on cytokine perturbations (IL-9, LT-alpha1-beta2) produces perturbation-specific shifts consistent with the true test distributions. (d) Generated vs. true cells across all cell types in the Replogle dataset. (e–f) For HepG2 cells, conditioning on genetic perturbations (PPP6C, ZDHHC7) yields realistic perturbation-dependent distributions that closely follow the experimental data.

In Appendix 19, we additionally report reconstruction results between the VAE component of scLDM and scVI for both datasets, showing how our improved transformer-based VAE significantly outperforms MLP-based scVI on the reconstruction task. Finally, we also tested how the additive conditioning for the classifier-free guidance proposed in Palma et al. (2025a) performs compared to the standard classifier-free guidance approach (Ho & Salimans, 2022). In Supplementary Table 19, we report that the standard approach is superior to the additive approach in multi-attribute conditional settings for perturbational single-cell data.

### 4.3 SCLDM-VAE EMBEDDING EVALUATIONS ON CLASSIFICATION TASKS

**Details** For the third experiment, we leveraged two datasets: the first dataset is a human lung single-cell RNA-sequencing data from healthy donors and patients affected by COVID-19 (Wu et al., 2024), the second dataset consists of 6 tissues from the Tabula Sapiens 2.0 (Consortium & Quake, 2025). The goal of this experiment is to verify the quality of embeddings provided by the auto-encoder on a downstream task (here: classification). We compare our approach to embeddings provided by TranscriptFormer (Pearce et al., 2025), scVI (Lopez et al., 2018), AIDO.Cell (Ho et al., 2024), Geneformer (Theodoris et al., 2023), scGPT (Cui et al., 2024), UCE (Rosen et al., 2023). We used Human Census data (CellxGene)[3] to train three versions of scLDM-VAE, namely, with around 20M parameters, 70M parameters, and 270M parameters. For our scLDM-VAE and benchmark models, both datasets represent out-of-distribution data that were unseen during training.

---

[3] https://cellxgene.cziscience.com/

To evaluate the quality of the learned representations, we process each of the four COVID-19 donors through the models to generate cell embeddings. For scLDM variants, we use the mean of the latent distribution, $\mu(\mathbf{x})$, which is flattened to a 4096-dimensional vector. To ensure fair comparison across models with different embedding dimensions, we apply principal component analysis (PCA) to all embeddings, retaining the top 128 principal components. For each donor independently, we train an unregularized logistic regression classifier to distinguish infected from uninfected cells using 5-fold cross-validation. The final metrics are computed as equally weighted averages across the four donors, with uncertainties propagated using standard error addition in quadrature: $\sigma_{\text{combined}} = \frac{1}{n} \sqrt{\sum_{i=1}^{n} \sigma_i^2}$, where $n = 4$ donors.

Table 4: COVID-19 model performance comparison (averaged across all donors). Since all standard errors are below 0.003 (see Appendix K.6), they are ommitted in this table.

| Model | F1 Score | Recall | Precision |
|---|---|---|---|
| TranscriptFormer | 0.814 | 0.829 | 0.801 |
| UCE | 0.775 | 0.781 | 0.771 |
| scGPT | 0.779 | 0.793 | 0.766 |
| Geneformer | 0.768 | 0.781 | 0.757 |
| AIDO.Cell | 0.717 | 0.729 | 0.708 |
| scVI | 0.675 | 0.680 | 0.680 |
| scLDM (20M) | 0.811 | 0.827 | 0.797 |
| scLDM (70M) | 0.815 | 0.830 | 0.801 |
| scLDM (270M) | **0.820** | **0.836** | **0.806** |

For the Tabula Sapiens 2.0 dataset, we evaluated cell type classification across 6 tissues: blood, spleen, lymph node, small intestine, thymus, and liver. Following the same protocol as the COVID-19 analysis, we stratified samples by tissue instead of donor and filtered out cell types with fewer than 100 cells to ensure robust classification. We employed multinomial logistic regression for the multi-class cell type prediction task. Final metrics are averaged over tissues with propagated uncertianites (see Appendix K.6).

**Results and discussion** As shown in Table 4, our 270M and 70M models achieve superior performance across all evaluated metrics for COVID-19 infection detection. The performance differences between scLDM (270M) and TranscriptFormer—the strongest benchmark model—represent meaningful differences given the measurement uncertainty, with our model achieving F1 score of $0.820 \pm 0.001$ compared to TranscriptFormer's $0.814 \pm 0.002$. The strong discriminative performance demonstrates that our transformer-based VAE learns biologically meaningful representations that capture infection-related transcriptional signatures. We observe substantial improvements over the VAE-based scVI model (F1: $0.675 \pm 0.001$), highlighting the advantages of our architectural innovations and model scale.

Table 5: Tabula Sapiens 2.0 model performance comparison (averaged across all tissues). Since all standard errors are below 0.003 (see Appendix K.6), they are ommitted in this table.

| Model | F1 Score | Recall | Precision |
|---|---|---|---|
| scGPT | 0.8 | 0.802 | 0.806 |
| scVI | 0.799 | 0.794 | 0.814 |
| TranscriptFormer | 0.799 | 0.8 | 0.802 |
| UCE | 0.796 | 0.797 | 0.801 |
| Geneformer | 0.777 | 0.776 | 0.786 |
| AIDO.Cell | 0.724 | 0.715 | 0.748 |
| scLDM (20M) | **0.804** | **0.805** | **0.812** |
| scLDM (70M) | 0.802 | 0.802 | 0.810 |
| scLDM (270M) | 0.802 | 0.803 | 0.811 |

For the Tabula Sapiens 2.0 classification results shown in Table 5, the differences in F1 scores between the scLDM model variants are within measurement uncertainty and may not be significant. Moreover, all top-performing models—scLDM variants, scGPT, scVI, and TranscriptFormer—achieve F1 scores within each other's uncertainties (ranging from 0.799 to 0.804 with standard errors of 0.002), indicating comparable performance for multi-class cell type classification. The consistent performance across both binary (COVID-19 infection) and multi-class (cell type) classification tasks validates the biological utility of our learned embeddings, making them valuable for biological discovery applications beyond generation.

## 5 CONCLUSION

In this paper, we demonstrate that enforcing the inductive bias of exchangeability is crucial for the generative modeling of single-cell data. We introduced a scalable architecture that combines a permutation-invariant encoder and a permutation-equivariant decoder within a fully transformer-

based VAE with a latent diffusion model parameterized using DiTs, achieving state-of-the-art performance on cell generation benchmarks, both observational and perturbational data, as well as downstream classification tasks. Our work extends beyond imposing artificial structure on gene expression data, instead providing a principled framework for learning from unordered sets. This approach is not limited to transcriptomics and lays the groundwork for developing foundational models for other exchangeable biological data, such as proteomics and epigenomics, as well as multi-omics and multi-modal data, thereby enabling more faithful and powerful virtual models of cellular biology.

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
