## A   LLM USAGE DISCLAIMER

We used large language models (ChatGPT and Claude) to assist with manuscript polishing, including grammar and clarity improvements, and to verify technical definitions and terminology. All scientific content, analysis, and conclusions are the original work of the authors.

## B   DEFINITIONS

**Self-attention**   Attention mechanism is defined as $\text{Att}(\mathbf{Q}, \mathbf{K}, \mathbf{V}) \overset{df}{=} \text{softmax}(\mathbf{Q}\mathbf{K}^\top)\mathbf{V}$, where $\mathbf{Q} \in \mathbb{R}^{m \times D}, \mathbf{K} \in \mathbb{R}^{M \times D}, \mathbf{V} \in \mathbb{R}^{M \times D}$, and $\text{softmax}(\cdot)$ is the row-wise softmax function.[4] Typically, $\mathbf{Q}, \mathbf{K}$ and $\mathbf{V}$ are results of linear transformations of given inputs. If all matrices are calculated based on the same input $\mathbf{X}$, we refer to it as *self-attention*.

**Cross-attention**   However, if $\mathbf{Q}$ is calculated using $\mathbf{X}$, and another input $\mathbf{Y}$ is used to obtain $\mathbf{K}$ and $\mathbf{V}$, then we refer to it as *cross-attention*.

**Multi-head attention**   $\text{Att}_K$ denotes a multi-head attention with $K$ heads, i.e., a concatenation of $K$ attention layers.

**Multi-head Attention Block (MAB)**   A multi-head attention block (MAB) is defined as follows Lee et al. (2019); Zhang et al. (2022):

$$\text{MAB}(\mathbf{X}, \mathbf{Y}) \overset{df}{=} f(\mathbf{X}, \mathbf{Y}) + \text{relu}(f(\mathbf{X}, \mathbf{Y})\mathbf{W}), \tag{9}$$

$$f(\mathbf{X}, \mathbf{Y}) \overset{df}{=} \mathbf{X}\mathbf{W}_f + \text{Att}_K(\mathbf{Q}(\mathbf{X}), \mathbf{K}(\mathbf{Y}), \mathbf{V}(\mathbf{Y})),$$

where $f(\mathbf{X}, \mathbf{Y}) \overset{df}{=} \mathbf{X}\mathbf{W}_f + \text{Att}_K(\mathbf{Q}(\mathbf{X}), \mathbf{K}(\mathbf{Y}), \mathbf{V}(\mathbf{Y}))$, $\mathbf{W}$ and $\mathbf{W}_f$ are learnable weight matrices.

**Set Attention Block (SAB)**   Set Attention Block (SAB) is defined as follows Lee et al. (2019):

$$\text{SAB}(\mathbf{X}) \overset{df}{=} \text{MAB}(\mathbf{X}, \mathbf{X}). \tag{10}$$

**Induced Set Attention Block (ISAB)**   Induced Set Attention Block (ISAB) is defined as follows Lee et al. (2019):

$$\text{ISAB}(\mathbf{X}) \overset{df}{=} \text{MAB}\left(\mathbf{X}, \text{MAB}(\mathbf{U}, \mathbf{X})\right), \tag{11}$$

where $\mathbf{U} \in \mathbb{R}^{m \times D}$ are *inducing points*, i.e., a global weight matrix learnable by backpropagation.

**Pooling by Multi-Head Attention**   Pooling by Multi-head Attention (PMA) is defined as follows:

$$\text{PMA}(\mathbf{X}) = \text{MAB}(\mathbf{S}, \text{rFF}(\mathbf{X})), \tag{12}$$

where $\mathbf{S} \in \mathbb{R}^{m \times D}$ is a matrix of learnable *inducing points* (or *pseudoinputs*), and $\text{rFF} : \mathbb{R}^{M \times D} \to \mathbb{R}^{M \times D}$ is a row-wise linear layer. For fixed inducing points $\mathbf{S}$ in PMA, this layer is permutation-invariant (this is due to applying the attention layer, see Property 3 in Appendix C).

## C   PERMUTATION-EQUIVARIANCE AND PERMUTATION-INVARIANCE OF ATTENTION MECHANISM

The row-wise self-attention function fulfills the following property:

**Property 1.** *For a given matrix* $\mathbf{X} \in \mathbb{R}^{M \times D}$, *and two permutation matrices* $\mathbf{P}_M \in \{0, 1\}^{M \times M}$ *and* $\mathbf{P}_D \in \{0, 1\}^{D \times D}$, *the following statements hold true for the row-wise softmax function: (i)* $\text{softmax}(\mathbf{X}\mathbf{P}_D^\top) = \text{softmax}(\mathbf{X})\mathbf{P}_D^\top$, *(ii)* $\text{softmax}(\mathbf{P}_M\mathbf{X}) = \mathbf{P}_M\text{softmax}(\mathbf{X})$.

---

[4]We skip scaling $\mathbf{Q}\mathbf{K}^\top$ by $1/\sqrt{D}$ to avoid unnecessary clutter.

Property 1 tells us that applying the permutation to the softmax function just reorders its columns/rows, hence, applying softmax before or after the reordering gives the same vector, merely shuffled.

Before we move to next properties, we recall that any permutation matrix is orthogonal, hence, $\mathbf{P}\mathbf{P}^\top = \mathbf{P}^\top\mathbf{P} = \mathbf{I}$.

It is a well-known fact that the (self-)attention mechanism is permutation-equivariant, namely, the following property holds true:

**Property 2.** *For a given permutation matrix* $\mathbf{P} \in \{0,1\}^{M \times M}$, *the attention mechanism is permutation-equivariant, i.e.,* $\mathrm{Att}\,(\mathbf{PQ}(\mathbf{X}), \mathbf{PK}(\mathbf{X}), \mathbf{PV}(\mathbf{X})) = \mathbf{P}\,\mathrm{Att}\,(\mathbf{Q}(\mathbf{X}), \mathbf{K}(\mathbf{X}), \mathbf{V}(\mathbf{X}))$.

*Proof.* First, we notice that: $\mathbf{Q}(\mathbf{PX}) = \mathbf{PXW}_Q = \mathbf{PQ}(\mathbf{X})$, $\mathbf{K}(\mathbf{PX}) = \mathbf{PXW}_K = \mathbf{PK}(\mathbf{X})$ and $\mathbf{V}(\mathbf{PX}) = \mathbf{PXW}_V = \mathbf{PV}(\mathbf{X})$. Next, to avoid unnecessary clutter, let us skip the dependency on $\mathbf{X}$. Then:

$$\mathrm{Att}\,(\mathbf{PQ}, \mathbf{PK}, \mathbf{PV}) = \mathrm{softmax}(\mathbf{PQ}(\mathbf{PK})^\top)\mathbf{PV}$$
$$= \mathrm{softmax}(\mathbf{PQK}^\top\mathbf{P}^\top)\mathbf{PV}$$
$$= \mathbf{P}\,\mathrm{softmax}(\mathbf{QK}^\top)\mathbf{P}^\top\mathbf{PV}$$
$$= \mathbf{P}\,\mathrm{softmax}(\mathbf{QK}^\top)\mathbf{V}$$
$$= \mathbf{P}\,\mathrm{Att}\,(\mathbf{Q}, \mathbf{K}, \mathbf{V})\,.$$

$\square$

The attention mechanism becomes permutation-invariant for $\mathbf{Q}$ being a global parameter matrix only if the following property holds true:

**Property 3.** *For a given permutation matrix* $\mathbf{P} \in \{0,1\}^{M \times M}$, *the attention mechanism with inducing points* $\mathbf{Q} \in \mathbb{R}^{m \times D}$ *is permutation-invariant, i.e.,* $\mathrm{Att}\,(\mathbf{Q}, \mathbf{PK}(\mathbf{X}), \mathbf{PV}(\mathbf{X})) = \mathrm{Att}\,(\mathbf{Q}, \mathbf{K}(\mathbf{X}), \mathbf{V}(\mathbf{X}))$.

*Proof.* First, we notice that: $\mathbf{K}(\mathbf{PX}) = \mathbf{PXW}_K = \mathbf{PK}(\mathbf{X})$ and $\mathbf{V}(\mathbf{PX}) = \mathbf{PXW}_V = \mathbf{PV}(\mathbf{X})$. Next, to avoid unnecessary clutter, let us skip the dependency on $\mathbf{X}$. Then:

$$\mathrm{Att}\,(\mathbf{Q}, \mathbf{PK}, \mathbf{PV}) = \mathrm{softmax}(\mathbf{Q}(\mathbf{PK})^\top)\mathbf{PV}$$
$$= \mathrm{softmax}(\mathbf{QK}^\top\mathbf{P}^\top)\mathbf{PV}$$
$$= \mathrm{softmax}(\mathbf{QK}^\top)\mathbf{P}^\top\mathbf{PV}$$
$$= \mathrm{softmax}(\mathbf{QK}^\top)\mathbf{V}$$
$$= \mathrm{Att}\,(\mathbf{Q}, \mathbf{K}, \mathbf{V})\,.$$

$\square$

However, for a latent matrix $\mathbf{Z} \in \mathbb{R}^{m \times D}$ obtained by transforming $\mathbf{X}$ in the permutation-invariant manner, an embedding matrix $\mathbf{E} \in \mathbb{R}^{V \times D}$, the inducing points determined by a set of indices $\mathcal{I}$, i.e., $\mathbf{Q} = \mathbf{E}_\mathcal{I}$, is permutation-equivariant if the indices $\mathcal{I}$ are permuted, i.e., a permutation of indices $\pi(\mathcal{I})$ induces the matrix $\mathbf{P}$, thus, $\mathbf{PQ} = \mathbf{E}_{\pi(\mathcal{I})}$. Then, the following property holds true:

**Property 4.** *For a given permutation of indices* $\mathcal{I}$, $\pi(\mathcal{I})$, *or, equivalently, a matrix permutation* $\mathbf{P} \in \{0,1\}^{|\mathcal{I}| \times |\mathcal{I}|}$, *and latents* $\mathbf{Z}$ *calculated by a permutation-invariant function* $f$, *i.e.,* $\mathbf{Z} = f(\mathbf{PX})$, *the attention mechanism with inducing points* $\mathbf{Q} \in \mathbb{R}^{|\mathcal{I}| \times D}$ *is permutation-equivariant, i.e.,* $\mathrm{Att}\,(\mathbf{PQ}, \mathbf{K}(f(\mathbf{PX})), \mathbf{V}(f(\mathbf{PX}))) = \mathbf{P}\mathrm{Att}\,(\mathbf{Q}, \mathbf{K}(\mathbf{Z}), \mathbf{V}(\mathbf{Z}))$.

*Proof.* First, since $f$ is permutation-invariance, we get $\mathbf{Z} = f(\mathbf{PX}) = f(\mathbf{X})$. Second, we note that $\mathbf{K}(\mathbf{PZ}) = \mathbf{PZW}_K = \mathbf{PK}(\mathbf{Z})$ and $\mathbf{V}(\mathbf{PZ}) = \mathbf{PZW}_V = \mathbf{PV}(\mathbf{Z})$. Then:

$$\mathrm{Att}\,(\mathbf{PQ}, \mathbf{K}(f(\mathbf{PX})), \mathbf{V}(f(\mathbf{PX}))) = \mathrm{softmax}(\mathbf{PQK}(f(\mathbf{PX}))^\top)\mathbf{V}(f(\mathbf{PX}))$$
$$= \mathbf{P}\,\mathrm{softmax}(\mathbf{QK}(f(\mathbf{X}))^\top)\mathbf{V}(f(\mathbf{X}))$$
$$= \mathbf{P}\,\mathrm{softmax}(\mathbf{QK}(\mathbf{Z})^\top)\mathbf{V}(\mathbf{Z})$$
$$= \mathbf{P}\,\mathrm{Att}\,(\mathbf{Q}, \mathbf{K}(\mathbf{Z}), \mathbf{V}(\mathbf{Z}))\,.$$

$\square$

# D  RELATED WORK (EXTENDED)

Modeling a probability distribution over order-agnostic objects like sets is challenging for at least two reasons. First, a model must be permutation-equivariant, meaning, changing the order of variables changes the order of parameters as well. Second, the model must also be exchangeable. Additionally, in the case of gene expression, for various tissues, we get different subsets of genes, thus, ideally, we would like to learn a single model to *transfer* hidden dependencies (correlations) among cells from distinct tissues.

**Autoregressive Models**  A varying-size objects are typically modeled by autoregressive models (ARMs) like transformer-based LLMs for text (Vaswani et al., 2017) or WaveNet for audio (Van Den Oord et al., 2016). However, ARMs assume a fixed order of variables, otherwise, like in the case of sets, their performance can drop significantly. Recently, it has been shown that misspecifying the order in ARMs can result in a huge drop in their performance (Kim et al., 2025). There are ways of dealing with the order in ARMs (Pannatier et al., 2024), but they are not well-suited for processing objects without an explicitly defined order.

**Masked Diffusion Models**  Recently, a masked version of diffusion-based models (Ho et al., 2020) are used to generate text quite successfully (Nie et al., 2025) since they can alleviate the need of specifying the order of generation. However, as proven in (Kim et al., 2025), masked diffusion-based models are order-agnostic but at the price of learning an extremely complex task of predicting a variable value conditioned on a set of unmasked variables in arbitrary positions.

**Variational Auto-Encoders**  Another modeling approach is to define a Variational Auto-Encoder (Kingma & Welling, 2014; Rezende et al., 2014) since this framework allows defining its components in a flexible manner. In (Kim et al., 2021), a SetVAE was formulated by introducing two separate latent variables to deal with varying size of sets, namely, $\mathbf{z}_{\mathcal{I}}$ of the same dimensionality as $\mathbf{x}_{\mathcal{I}}$ such that $\mathbf{z}_i$ corresponds to $\mathbf{x}_i$, $i \in \mathcal{I}$, and an additional vector of latents $\mathbf{c} \in \mathbb{R}^{d_1}$ of a constant size $d_1$. In general, $\mathbf{x}_{\mathcal{I}}$ can be generated given $\mathbf{z}_{\mathcal{I}}$ and each $\mathbf{z}_i$ is generated given $\mathbf{c}$, i.e., $p(\mathbf{x}_{\mathcal{I}}|\mathcal{I}, \mathbf{z}_{\mathcal{I}}) \, p(\mathbf{z}_{\mathcal{I}}|\mathbf{c}) \, p(\mathbf{c})$, where $p(\mathbf{x}_{\mathcal{I}}|\mathcal{I}, \mathbf{z}_{\mathcal{I}}) = \prod_{i=1}^{|\mathcal{I}|} p(\mathbf{x}_i|\mathbf{z}_i)$ and $p(\mathbf{z}_{\mathcal{I}}|\mathbf{c}) = p(|\mathcal{I}|) \prod_{i=1}^{|\mathcal{I}|} p(\mathbf{z}_i|\mathbf{c})$. Then, the variational posteriors can take the following form: $q(\mathbf{z}_{\mathcal{I}}|\mathbf{x}_{\mathcal{I}}) = \delta(|\mathcal{I}|) \prod_{i=1}^{|\mathcal{I}|} q(\mathbf{z}_i|\mathbf{x}_i)$, where $\delta(\cdot)$ is Dirac's delta, and additionally we have $q(\mathbf{c}|\mathbf{x}_{\mathcal{I}})$. In (Kim et al., 2021), a few simplifications were made such that the model fits well modeling point clouds (sets of 3-D points), namely, for all $i = 1, \ldots, |\mathcal{I}|$, $p(\mathbf{z}_i|\mathbf{c}) = p(\mathbf{z}_i)$, and $q(\mathbf{z}_i|\mathbf{c}) = p(\mathbf{z}_i)$. Further, the authors of (Kim et al., 2021) suggested to define a hierarchical VAE with multiple layers of $\mathbf{z}_{\mathcal{I}}$'s and $\mathbf{c}$'s since a single layer did not result in good performance, and they replaced the conditional likelihood with Chamfer Distance as a well-suited distance for point clouds. In this paper, we find a great appeal of the VAE framework and its flexibility; however, we claim using two distinct latents and a hierarchical latent structure to be unnecessary. Instead, we suggest picking a careful parameterization to be *crucial* in obtaining high performance.

**Permutation-equivariant/invariant Parameterizations**  Deep Neural Networks are widely used as transformations of raw data and parameterizations of probability distributions. It is advocated (but also observed empirically) that modeling probability distributions requires utilizing symmetries in data (Bronstein et al., 2021). For instance, for objects whose dimensions can be shuffled without changing the underlying latent structure, we need either *permutation-invariant* or *permutation-equivariant* transformations. For a given permutation matrix $\mathbf{P}$, a function $f : \mathbb{X} \to \mathbb{Y}$ is *permutation-invariant* if $f(\mathbf{P}\mathbf{x}) = \mathbf{y}$; on the other hand, a function $f : \mathbb{X} \to \mathbb{Y}$ is *permutation-equivariant* if $f(\mathbf{P}\mathbf{x}) = \mathbf{P}\mathbf{y}$.

A general *blueprint* for composing geometric deep neural networks is a composition of permutation-invariant layers and/pr permutation-equivariant layers, with nonlinearity activation functions in between (Bronstein et al., 2021). An example of such a blueprint is an architecture called DeepSets (Zaheer et al., 2017). It formulates a general permutation-equivariance layer treating all variables consistently regardless their positions. Then it applies a symmetric aggregation like averaging or other pooling operators (Kimura et al., 2024; Ilse et al., 2018; Xie & Tong, 2025) to combine these

equivariant features in a permutation-invariant fashion, ensuring the order of inputs does not affect the output. The drawback of this approach is that all elements are processed separately before being aggregated with a non-learnable pooling operator. This manner of constructing permutation-invariant transformations might be highly limiting.

A potential solution to that issue is replacing static pooling with a learned attention mechanism, allowing utilizing transformer-based architectures to transform all elements jointly in an equivariant and invariant fashion. An example of a fully-transformer-based model is SetTransformer (Lee et al., 2019) that builds on the following idea. Attention mechanism is defined as $\text{Att}(\mathbf{Q}, \mathbf{K}, \mathbf{V}) \overset{df}{=} \text{softmax}(\mathbf{Q}\mathbf{K}^\top)\mathbf{V}$, where $\mathbf{Q} \in \mathbb{R}^{m \times D}, \mathbf{K} \in \mathbb{R}^{M \times D}, \mathbf{V} \in \mathbb{R}^{M \times D}$, and $\text{softmax}(\cdot)$ is the row-wise softmax function.[5] Typically, $\mathbf{Q}$, $\mathbf{K}$ and $\mathbf{V}$ are results of linear transformations of some inputs. If all matrices are calculated based on the same input $\mathbf{X}$, we refer to it as *self-attention*. However, if $\mathbf{Q}$ is calculated using $\mathbf{X}$, and another input $\mathbf{Y}$ is used to obtain $\mathbf{K}$ and $\mathbf{V}$, then we refer to it as *cross-attention*. Let $\text{Att}_K$ denote a multi-head attention with $K$ heads, i.e., a concatenation of $K$ attention layers. SetTransformer introduces a multi-head attention block (MAB) Lee et al. (2019); Zhang et al. (2022):

$$\text{MAB}(\mathbf{X}, \mathbf{Y}) \overset{df}{=} f(\mathbf{X}, \mathbf{Y}) + \text{relu}(f(\mathbf{X}, \mathbf{Y})\mathbf{W}), \tag{13}$$

$$f(\mathbf{X}, \mathbf{Y}) \overset{df}{=} \mathbf{X}\mathbf{W}_f + \text{Att}_K(\mathbf{Q}(\mathbf{X}), \mathbf{K}(\mathbf{Y}), \mathbf{V}(\mathbf{Y})),$$

where $f(\mathbf{X}, \mathbf{Y}) \overset{df}{=} \mathbf{X}\mathbf{W}_f + \text{Att}_K(\mathbf{Q}(\mathbf{X}), \mathbf{K}(\mathbf{Y}), \mathbf{V}(\mathbf{Y}))$, $\mathbf{W}$ and $\mathbf{W}_f$ are learnable weight matrices. Given MAB, SetTransformer further defines the following two blocks, namely, the Set Attention Block (SAB) and Induced Set Attention Block (ISAB): $\text{SAB}(\mathbf{X}) \overset{df}{=} \text{MAB}(\mathbf{X}, \mathbf{X})$, and $\text{ISAB}(\mathbf{X}) \overset{df}{=} \text{MAB}(\mathbf{X}, \text{MAB}(\mathbf{U}, \mathbf{X}))$, where $\mathbf{U} \in \mathbb{R}^{m \times D}$ are *inducing points*, i.e., a global weight matrix learnable by backpropagation. ISAB allows to change the size of the input, and similarly to SAB, it is permutation-equivariant (Lee et al., 2019). To obtain a permutation-invariant transformation, SetTransformer proposes to use another layer called Pooling by Multi-head Attention (PMA):

$$\text{PMA}(\mathbf{X}) = \text{MAB}(\mathbf{S}, \text{rFF}(\mathbf{X})), \tag{14}$$

where $\mathbf{S} \in \mathbb{R}^{m \times D}$ is a matrix of learnable *inducing points* (or *pseudoinputs*), and $\text{rFF} : \mathbb{R}^{M \times D} \to \mathbb{R}^{M \times D}$ is a row-wise linear layer. For fixed inducing points $\mathbf{S}$ in PMA, this layer is permutation-invariant (this is due to applying the attention layer, see Property 3 in Appendix C). These building blocks can be used to formulate a deep neural network for parameterizing a probabilistic model. However, we advocate for a different parameterization that applies a single multi-head attention layer in a transformer block to obtain a fixed-size output, and then a series of transformer blocks.

**Latent Diffusion Models** Latent Diffusion Models (LDMs) perform diffusion processes in learned latent spaces rather than directly in high-dimensional data spaces. Stable Diffusion (Rombach et al., 2022) pioneered this approach for text-to-image synthesis by training diffusion models in the latent space of a pre-trained VAE, dramatically reducing computational costs while maintaining generation quality. This paradigm has proven effective across diverse scientific domains: all-atom diffusion transformers (Joshi et al., 2025) generate molecules and materials with atomic-level precision, similary LaM-SLidE (Sestak et al., 2025) utilizes transformer-based LMD for molecular dynamics (among others), while La-proteina (Geffner et al., 2025) employs transformer-based partially latent flow matching for atomistic protein generation. These advances demonstrate the versatility of latent diffusion approaches for complex, high-dimensional scientific data across multiple modalities. Here, we extend this framework to single-cell transcriptomics by proposing a transformer-based LDM for this biological data type.

**Generative Models for scRNA-seq** In the context of single-cell genomics, numerous generative models have been developed for (conditional) sampling of gene expression profiles. scVI (Lopez et al., 2018) represents an early VAE-based generative model, while more recent approaches include GAN-based and diffusion-based architectures such as scGAN (Marouf et al., 2020a) and scDiffusion (Luo et al., 2024). These models operate in continuous space and therefore transform discrete gene expression data into log-normalized counts. Recently, latent diffusion frameworks have emerged with models like SCLD (Wang et al., 2023) and CFGen (Palma et al., 2025a), which leverage latent diffusion frameworks. Additionally, application-specific generative models have been

---

[5]We skip scaling $\mathbf{Q}\mathbf{K}^\top$ by $1/\sqrt{D}$ to avoid unnecessary clutter.

developed for perturbational single-cell genomics, including CPA Lotfollahi et al. (2023a), SquiD-iff (He et al., 2024), CellFlow (Klein et al., 2025), and CellOT (Bunne et al., 2023), which are tailored to capture the effects of genetic and chemical perturbations on cellular states. Our approach is similar in vein to CFGen and SCLD, but leverages transformer-based architectures for both our newly proposed VAE as well as the latent diffusion model.

# E  OUR APPROACH: ADDITIONAL INFORMATION

## E.1  GENE EXPRESSION DATA EMBEDDING: REPLACING *dropouts*

We present a method for processing sparse gene expression data that focuses computational resources on biologically relevant signals. Given a set of $D$ genes with their corresponding expression counts, our approach addresses the inherent sparsity in single-cell RNA sequencing data, where typically 70% or more of gene-cell entries are zero.

Let $\mathcal{I} = \{1, 2, \ldots, D\}$ denote the complete set of gene IDs represented as integers, and let $\mathbf{x} = (x_1, x_2, \ldots, x_D)$ represent the corresponding gene expression counts for a given cell, where $x_i \in \mathbb{N}_0$ is the count for gene $g_i$, then an $n$-th single cell is defined as a tuple $(\mathbf{x}_{\mathcal{I}_n}, \mathcal{I}_n)$.

Our method proceeds as follows:

1. **Context length constraint**: We define a maximum context length $d < D$ to limit the computational complexity of downstream processing.

2. **Expression-based filtering**: For each cell, we identify the set of expressed genes:
$$\mathcal{E} = \{i \in \mathcal{I} : x_i > 0\} \tag{15}$$
with corresponding expression values $\mathbf{x}_{\mathcal{E}} = \{x_i : i \in \mathcal{E}\}$.

3. **Context construction**: We construct a fixed-length input representation of dimension $d$. When $|\mathcal{E}| < d$ (which is typically the case due to high sparsity), we pad the input with artificial tokens to maintain consistent dimensionality:
$$\text{Input} = \begin{cases} \{(x_i, i)\}_{i \in \mathcal{E}} \cup \{(0, \text{PAD})\}^{d-|\mathcal{E}|} & \text{if } |\mathcal{E}| < d \\ \{(x_i, i)\}_{i \in \mathcal{E}} & \text{if } |\mathcal{E}| = d \end{cases} \tag{16}$$
where PAD is a special token for zero expression count.

This approach offers both computational and biological advantages. By excluding zero-expression genes (dropouts) from the input representation, we enable the model to focus exclusively on expressed genes, which carry the meaningful biological signal. The padding tokens serve purely as placeholders for implementation consistency and do not introduce spurious biological information, as they are explicitly marked with zero counts. This design choice aligns with the biological understanding that in single-cell data, the absence of detected expression often represents technical dropouts rather than meaningful biological zeros, making it advantageous to direct the model's attention solely to the detected expression events.

## E.2  CONDITIONAL LIKELIHOOD: THE PARAMETERIZATION OF NEGATIVE BINOMIAL

We model the gene expression counts using a Negative Binomial distribution, which effectively captures the overdispersion commonly observed in single-cell RNA-seq data. The conditional likelihood for our model is specified as follows.

Let $\mathbf{h}(\mathbf{Z}) \in \mathbb{R}^D$ denote the output of our neural network for a given cell embeddibg $\mathbf{Z}$, where $D$ is the number of genes. We apply a softmax transformation to obtain normalized ratios:

$$p_i(\mathbf{Z}) = \frac{\exp(\mathbf{h}_i(\mathbf{Z}))}{\sum_{j=1}^{D} \exp(\mathbf{h}_j(\mathbf{Z}))} \tag{17}$$

where $i = 1, 2, \ldots, D$, and $\sum_{i=1}^{D} p_i = 1$.

To obtain the expected expression counts, we scale these probabilities by the cell-specific library size $L$, namely:

$$\eta_i(\mathbf{Z}) = L \cdot p_i(\mathbf{Z}) \tag{18}$$

where $\mu_i$ represents the mean parameter for gene $i$ in the Negative Binomial distribution.

The gene expression count $x_i$ for gene $i$ is then modeled as:

$$x_i \sim \text{NB}(\eta_i(\mathbf{Z}), \alpha_i) \tag{19}$$

where NB denotes the Negative Binomial distribution parameterized by mean $\mu_i$ and dispersion $\alpha_i$. The probability mass function is given by:

$$p(x_i | \eta_i(\mathbf{Z}), \alpha_i) = \frac{\Gamma(x_i + \alpha_i^{-1})}{\Gamma(x_i + 1)\Gamma(\alpha_i^{-1})} \left( \frac{\alpha_i^{-1}}{\alpha_i^{-1} + \eta_i(\mathbf{Z})} \right)^{\alpha_i^{-1}} \left( \frac{\eta_i(\mathbf{Z})}{\alpha_i^{-1} + \eta_i(\mathbf{Z})} \right)^{x_i} \tag{20}$$

We consider two parameterizations for the dispersion:

1. **Shared dispersion**: A single parameter $\alpha$ is used for all genes, i.e., $\alpha_i = \alpha$ for all $i \in \{1, 2, \ldots, D\}$. This reduces the number of parameters and assumes homogeneous overdispersion across genes.

2. **Gene-specific dispersion**: Each gene has its own dispersion parameter, resulting in a vector $\boldsymbol{\alpha} = (\alpha_1, \alpha_2, \ldots, \alpha_D)$. This allows for heterogeneous overdispersion patterns across genes, providing greater flexibility at the cost of additional parameters.

This formulation ensures that the predicted expression values respect the constraint that total counts sum to the observed library size, while the Negative Binomial distribution appropriately models the count nature and overdispersion of the data. The softmax transformation guarantees that the neural network learns a proper distribution over genes, making the model interpretable as learning the relative expression probabilities for each cell.

## F  DATASETS

**General**  In our experiments, we used the following datasets: Dentate gyrus, Tabula Muris, Human Lung Census Atlas (HLCA), Parse1M and Replogle-Nadig; see Table 6 for details.

**Experiment 1: Cell generation (benchmarks)**  In the cell generation experiment, we used three widely used datasets, namely, Dentate gyrus, Tabula Muris, and HLCA. Dentate gyrus is the smallest dataset (only 18k cell and 17k genes). Tabula Muris is a small dataset with over 245k cells and almost 20k genes. Human Lung Cell Atlas (HLCA) is the largest, having about 585k cells and almost 28k genes.

**Experiment 2: Parse1M & Replogle**  In the second experiment, we used a curated subset of 10 Million Human PBMC dataset. We carried out experiments on 2k highly variable genes (HVGs). In this data, we focused on a single donor who had 18 cell types undergone 90 cytokine perturbations as well as a control treatment. We left out for testing the cell type 'CD4 Naive' and 27 cytokine perturbations.

Next, we used the well-known Replogle-Nadig dataset which consists of four cell lines and 2024 gene-edits. We carried out experiments on 2k highly variable genes (HVGs). All 2024 gene-edits in three cell-lines ('jurkat', 'k562', 'rpe1'), along with a subset of edits from the 'hepg2' cell line were used for taining. The remaining 'hepg2' gene-edits were held out for testing. We held out 372 gene edits from hepg2 cells.

**Experiment 3: COVID-19 and Tabula Sapiens 2.0**  In the fourth experiment, we used two datasets for embedding evaluation. First, we used the scRNA-seq experimental dataset of four healthy donors' lung sections infected with SARS-CoV-2 (Wu et al., 2024). Data were downloaded from CZ CELLxGENE[6]. Second, we used the Tabula Sapiens 2.0 dataset (Consortium & Quake,

---

[6] https://cellxgene.cziscience.com/collections/2a9a17c9-1f61-4877-b384-b8cd5ffa4085

2025), a comprehensive single-cell atlas of human tissues. We focused on 6 tissues: blood, spleen, lymph node, small intestine, thymus, and liver. We filtered out cell types with fewer than 100 cells to ensure robust classification performance and used the resulting filtered dataset for multinomial logistic regression-based cell type prediction tasks.

Table 6: Summary of datasets used in the experiments.

| Experiment | Dataset name | No. of cells | No. of genes | No. of cell types/lines |
|---|---|---|---|---|
| 1 | Dentate gyrus | 18,213 | 17,002 | 14 cell types |
| 1 | Tabula Muris | 245,389 | 19,734 | 123 cell types |
| 1 | HLCA | 584,944 | 27,997 | 50 cell types |
| 2 | Parse1M | 1,267,690 | 2,000 (HVGs) | 18 cell types |
| 2 | Replogle-Nadig | 624,158 | 2,000 (HVGs) | 4 cell lines |
| 3 | COVID-19 | 354,026 | 27,998 | 55 cell types |
| 3 | Tabula Sapiens 2.0 | 1,482,026 | $\sim 25k$ | 22 cell types |

## G    BASELINES

**scVI**    Single-cell Variational Inference (scVI) (Lopez et al., 2018) is VAE-based generative models designed for single-cell discrete data. Following the standard VAE framework, this model learns a Gaussian latent space that is subsequently decoded into the parameters of a discrete conditional likelihood model. For the reconstruction. For the observational data experiments, we implemented our own scVI model following default parameters from (Palma et al., 2025a). For the perturbational dataset, we used the implementation from the State Adduri et al. (2025) reproducibility repo https://github.com/ArcInstitute/State-reproduce, using default parameters. We train the models for 120k steps.

**scDiffusion**    A version of a latent diffusion model for single-cell gene expression data is scDiffusion (Luo et al., 2024). The scDiffusion model consists of three modules. The first module is an auto-encoderthat transforms gene expression patterns into a compact representation space, allowing dimensionality reduction and identification of complex cellular measurements. In the latent space, a denoising network is trained to reverse a diffusion process applied to the latent embeddings, turning noise into meaningful biological signal encoded in the latent space. To ensure guided generation, a third model is trained, a classifier, for incorporating cell type or other biological attributes.

**CFGen**    CFGen is a current state-of-the-art latent diffusion model that builds upon scVI, training a latent flow matching model in the VAE's latent space (Palma et al., 2025a). Similar to our approach, CFGen employs a two-stage training strategy: first training the autoencoder, then training the flow matching model on the VAE-generated embeddings. While CFGen introduces additive steering through classifier-free guidance, we utilize joint attribute control (see Table 19). However, since in the observational experiments we only conditioned on a single attribute, we do not think this is the source of the performance difference. Additionally, CFGen models the library size within the diffusion framework and samples from the mean and standard deviation of the library size distribution for conditional generation. We adapted this approach for sampling library size in our Negative Binomial conditional likelihood; however, unlike CFGen, we do not condition our Diffusion Transformer model on library size. We did not retrain CFGen but used the checkpoints for each observational dataset from the original repository. We followed the notebook for sampling from the model using guidance value of 1 (default).

**CPA**    Compositional Perturbation Autoencoder (CPA) (Lotfollahi et al., 2023b) is a deep generative model developed to predict gene expression changes under perturbations and their combinations. CPA disentangles latent representations of basal cellular state, perturbation effects, and additional covariates such as cell type. By recombining these factors through its decoder, CPA can reconstruct observed expression profiles and generalize to unseen perturbation–covariate combinations. This compositional structure enables CPA to extrapolate beyond training data, making it particularly well-suited for evaluating out-of-distribution generalization in perturbational single-cell datasets. For each dataset (Replogle and Parse1M) we trained 3 models with different seeds, but the same

leave-out test set. We used the implementation from the State Adduri et al. (2025) reproducibility repo `https://github.com/ArcInstitute/State-reproduce`, using default parameters. We train the models for 120k steps.

**scGPT** For each dataset (Replogle and Parse1M) we trained 3 models with different seeds, but the same leave-out test set. We used as baseline the scGPT Cui et al. (2024) model in the 'genetic' configuration for the Replogle-Nadig dataset, and in the 'chemical' configuration for the Parse 1M dataset. We used the implementation from the State Adduri et al. (2025) reproducibility repo `https://github.com/ArcInstitute/State-reproduce`, using default parameters. We train both scGPT models for 80k steps. For the 'genetic' configuration, we noticed that not all genes were present in the vocabulary (scGPT human downloaded from the original repo, following the baseline reproducibility repo from STATE), and that some genetic perturbations would be dropped. Hence, we built a feature set that consists of the union of 2000 HVG as well as the 2024 genetic perturbations, recovering 3482 genes. The 'scGPT genetic' model was therefore trained on 3482 genes, but evaluated on 1962 genes, which correspond to the subset of HVG that are part of the scGPT vocabulary. For the 'chemical' configuration, we trained on the same feature set of the rest of the baselines 2000 HVG.

**STATE** For each dataset (Replogle and Parse1M) we trained 3 models with different seeds, but the same leave-out test set. We used the STATE-Tx implementation from `https://github.com/ArcInstitute/State-reproduce` and trained for 80k steps for both datasets. We used the model configurations used in this notebook `https://colab.research.google.com/drive/1Ih-KtTEsPqDQnjTh6etVv_f-gRAA86ZN` for both datasets.

**CellFlow** For each dataset (Replogle and Parse1M) we trained 3 models with different seeds, but the same leave-out test set. We used the CellFlow implementation with gene and condition embedding computed from the mean expression of the training data. We used PCA for encoding and reconstruction. We trained for 500k steps for both datasets. We used the model configurations used in this notebook `https://cellflow.readthedocs.io/en/latest/notebooks/100_pbmc.html` for both datasets.

**Perturb mean** The Perturbation Mean baseline provides predictions by combining cell-type-specific control means with learned perturbation effects. The model operates as follows:

For each cell type $c$ and perturbation $p$, we calculate separate means for control and perturbed populations:

$$\boldsymbol{\mu}_c^{\text{ctrl}} = \frac{1}{|\mathcal{C}_c|} \sum_{i \in \mathcal{C}_c} \mathbf{x}^{(i)}, \quad \boldsymbol{\mu}_{c,p}^{\text{pert}} = \frac{1}{|\mathcal{P}_{c,p}|} \sum_{i \in \mathcal{P}_{c,p}} \mathbf{x}^{(i)} \tag{21}$$

Here, $\mathcal{C}_c$ denotes the collection of control (ctrl) cells belonging to type $c$, while $\mathcal{P}_{c,p}$ represents perturbed cells of type $c$ that received perturbation $p$. The cell-type-specific perturbation effect is computed as:

$$\boldsymbol{\delta}_{c,p} = \boldsymbol{\mu}_{c,p}^{\text{pert}} - \boldsymbol{\mu}_c^{\text{ctrl}} \tag{22}$$

To obtain a global perturbation effect, we average these cell-type-specific effects across all cell types where the perturbation was applied:

$$\boldsymbol{\delta}_p = \frac{1}{|\mathcal{C}_p|} \sum_{c \in \mathcal{C}_p} \boldsymbol{\delta}_{c,p}, \quad \text{where } \mathcal{C}_p = \{c \mid |\mathcal{P}_{c,p}| > 0\} \tag{23}$$

For prediction, given a test cell of type $t$ and perturbation $p$, the model generates:

$$\hat{\mathbf{x}} = \boldsymbol{\mu}_t^{\text{ctrl}} + \boldsymbol{\delta}_p, \quad \boldsymbol{\delta}_{\text{ctrl}} = \mathbf{0} \tag{24}$$

This approach ensures that control cells are predicted without modification, while perturbed cells receive a consistent global perturbation shift regardless of their specific cell type.

**Hepg2 and CD4 mean, context mean** The Context Mean baseline predicts a cell's post-perturbation profile by returning the average perturbed expression of cells of the same cell type observed in the training set. For every cell type $c$, we collect all training cells whose perturbation is not the control and form the pseudo-bulk mean:

$$\boldsymbol{\mu}_c = \frac{1}{|\mathcal{T}_c|} \sum_{i \in \mathcal{T}_c} \mathbf{x}^{(i)}, \quad \mathcal{T}_c = \left\{ i \mid \text{cell\_type}(i) = c, \, p^{(i)} \neq \text{ctrl} \right\}. \tag{25}$$

At inference time, for a test cell $i$ with cell type $c^{(i)}$ and perturbation label $p^{(i)}$, we predict:

$$\hat{\mathbf{x}}^{(i)} = \begin{cases} \mathbf{x}^{(i)} & \text{if } p^{(i)} = \text{ctrl}, \\ \boldsymbol{\mu}_{c^{(i)}} & \text{if } p^{(i)} \neq \text{ctrl}. \end{cases} \tag{26}$$

In other words, control cells are passed through unchanged, whereas perturbed cells inherit their cell-type mean.

## H    IMPLEMENTATION DETAILS

In this paper, we carried out model selection for various values of hyperparameters. In the following paragraphs, we provide further details for reproducibility.

**VAE**    Table 7 summarizes the hyperparameter configurations used for the VAE encoder architectures in our experiments.

Table 7: Hyperparameter values of VAE Encoders considered in this paper.

| VAE Encoder | |
|---|---|
| **Embedding layer** | |
| Embedding size | 256 |
| **Cross-Attention Block** | |
| Number of Heads | $\{1,4,8\}$ |
| No. pseudoinputs | $\{64,128, 256\}$ |
| Embedding size | $\{128,256\}$ |
| **Transformer Blocks** | |
| Number of Blocks | $\{2, 4\}$ |
| Number of Heads | 1 |
| Embedding size | 256 |
| **Gaussian Stochastic Layer** | |
| Latents per token | $\{8, 16, 32\}$ |

Table 8 summarizes the hyperparameter configurations used for the VAE decoder architectures in our experiments.

**Flow Matching**    Table 9 summarizes the hyperparameter configurations used for the LDM architectures in our experiments.

**scLDM-VAE Census**    Table 10 summarizes the hyperparameter configurations used for the scLDM-VAE Census architectures in our experiments.

**Training details**    For training, we swept over various configurations of hyperparameters, see Table 11.

Table 8: Hyperparameter values of VAE Decoders considered in this paper.

| VAE Decoder | |
|---|---|
| **Transformer Blocks** | |
| Number of Blocks | {2, 4} |
| Number of Heads | {1,4,8} |
| Embedding size | {128,256} |
| Normalization | `LayerNorm` |
| **Cross-Attention Block** | |
| Shared embedding layer | `True` |
| No. pseudoinputs | {64,128, 256} |
| Number of Heads | 1 |
| Embedding size | 256 |
| **NegativeBinomial Stochastic Layer** | |
| Shared $\theta$ | `False` |

Table 9: Hyperparameter values of LDMs considered in this paper.

| LDM – Flow Matching | |
|---|---|
| **Denoising Transformer** | |
| Number of Blocks | 8 |
| Number of Heads | 8 |
| Embedding size | 256 |
| Normalization | `LayerNorm` |
| Adaptive Normalization | `True` |
| **Hyperparams** | |
| $\sigma$ | $1e^{-4}$ |
| $v$ | 0 |
| Transport | `linear` |

Table 10: Hyperparameter values of scLDM-VAE Census.

| Encoder-Decoder hyperparameters | |
|---|---|
| Number of Blocks | {8,16} |
| Number of Heads | {8,16} |
| Number of Layers | {8,16} |
| Number of Latent Tokens | {256} |
| Embedding size | {256,512,768} |
| Normalization | `LayerNorm` |

Table 11: Hyperparameter values of training procedures considered in this paper.

| Training | |
|---|---|
| KL-weight | $\{0, 1e^{-5}\}$ |
| Optimizer | AdamW |
| Mini-batch size | $\{64, 128, 256\}$ |
| Learning rate | $\{1e^{-3}, 5e^{-4}\}$ |
| $(\beta_1, \beta_2)$ | $(0.9, 0.95)$ |
| Weight Decay | $\{0, 1e^{-7}, 1e^{-4}\}$ |
| Learning scheduler | cosine |

**Model Complexity**  In Tables 12 and 13, we report the computational complexity and parameter counts for the models used in our experiments. Table 12 shows the FLOPs and parameter counts for both the VAE and LDM components across five different single-cell datasets: dentate gyrus, tabula muris, HLCA, Replogle, and Parse1M. Table 13 presents the FLOPs and parameter counts for three VAE models trained on the Census datasets, with model sizes ranging from 20M to 270M parameters.

Table 12: Model Complexity Comparison Across Datasets

| Dataset | VAE FLOPs | LDM FLOPs | VAE Params | LDM Params |
|---|---|---|---|---|
| dentate_gyrus | 5.02 GFLOPs | 6.45 GFLOPs | 3.45M | 9.77M |
| tabula_muris | 29.45 GFLOPs | 77.33 GFLOPs | 13.23M | 57.81M |
| hlca | 38.13 GFLOPs | 77.33 GFLOPs | 15.35M | 57.83M |
| replogle | 28.02 GFLOPs | 51.56 GFLOPs | 21.16M | 39.91M |
| parse1m | 28.02 GFLOPs | 51.56 GFLOPs | 21.16M | 38.93M |

Table 13: Model Complexity Comparison

| Model | VAE TFLOPs | VAE Params |
|---|---|---|
| VAE_Census_20M | 0.33 TFLOPs | 23.75M |
| VAE_Census_70M | 1.02 TFLOPs | 76.07M |
| VAE_Census_270M | 2.63 TFLOPs | 270.14M |

# I  EVALUATION

## I.1  MAXIMUM MEAN DISCREPANCY (MMD)

We propose to use the Maximum Mean Discrepancy (MMD) (Gretton et al., 2012). MMD is a non-parametric distance measure between probability distributions based on the notion of embedding distributions into a reproducing kernel Hilbert space (RKHS) $\mathcal{H}$. Given two distributions $P$ and $Q$ over a domain $\mathcal{X}$, the MMD is defined as:

$$\text{MMD}[\mathcal{F}, P, Q] = \sup_{f \in \mathcal{F}} \left( \mathbb{E}_{x \sim P}[f(x)] - \mathbb{E}_{y \sim Q}[f(y)] \right), \tag{27}$$

where $\mathcal{F}$ is a class of functions. When $\mathcal{F}$ is the unit ball in an RKHS $\mathcal{H}$ with kernel $k$, the MMD can be expressed as:

$$\text{MMD}^2[\mathcal{H}, P, Q] = \mathbb{E}_{x,x' \sim P}[k(x, x')] + \mathbb{E}_{y,y' \sim Q}[k(y, y')] - 2\mathbb{E}_{x \sim P, y \sim Q}[k(x, y)]. \tag{28}$$

In practice, given finite samples $X = \{x_1, \ldots, x_m\}$ drawn from $P$ and $Y = \{y_1, \ldots, y_n\}$ drawn from $Q$, we use the unbiased empirical estimate:

$$\widehat{\text{MMD}}^2[X, Y] = \frac{1}{m(m-1)} \sum_{i \neq j}^{m} k(x_i, x_j) + \frac{1}{n(n-1)} \sum_{i \neq j}^{n} k(y_i, y_j) - \frac{2}{mn} \sum_{i=1}^{m} \sum_{j=1}^{n} k(x_i, y_j). \quad (29)$$

The choice of kernel $k$ determines the richness of the function class $\mathcal{F}$. Common choices include the Gaussian RBF kernel $k(x, y) = \exp(-\|x - y\|^2/2\sigma^2)$ with bandwidth parameter $\sigma$. The MMD is zero if and only if $P = Q$ when using a characteristic kernel, making it a powerful tool for two-sample testing and distribution matching applications.

## I.2 2-WASSERSTEIN DISTANCE (W2)

The 2-Wasserstein distance provides an alternative metric for comparing probability distributions based on optimal transport theory. For distributions $P$ and $Q$ on $\mathbb{R}^d$, the 2-Wasserstein distance is defined as:

$$W_2(P, Q) = \left( \inf_{\gamma \in \Gamma(P,Q)} \int_{\mathbb{R}^d \times \mathbb{R}^d} \|\mathbf{x} - \mathbf{y}\|^2 \, d\gamma(\mathbf{x}, \mathbf{y}) \right)^{1/2}, \quad (30)$$

where $\Gamma(P, Q)$ denotes the set of all joint distributions with marginals $P$ and $Q$. For empirical distributions with equal sample sizes $n$, given samples $X = \{\mathbf{x}_1, \ldots, \mathbf{x}_n\}$ and $Y = \{\mathbf{y}_1, \ldots, \mathbf{y}_n\}$, the discrete 2-Wasserstein distance simplifies to:

$$W_2^2(X, Y) = \frac{1}{n} \min_{\pi \in \Pi_n} \sum_{i=1}^{n} \|\mathbf{x}_i - \mathbf{y}_{\pi(i)}\|^2, \quad (31)$$

where $\Pi_n$ is the set of all permutations of $\{1, \ldots, n\}$. This optimization problem can be solved efficiently using the Hungarian algorithm or entropic regularization approaches.

When both distributions are Gaussian with means $\boldsymbol{\mu}_P, \boldsymbol{\mu}_Q$ and covariances $\boldsymbol{\Sigma}_P, \boldsymbol{\Sigma}_Q$, the 2-Wasserstein distance has a closed-form expression:

$$W_2^2(P, Q) = \|\boldsymbol{\mu}_P - \boldsymbol{\mu}_Q\|^2 + \text{tr}\left( \boldsymbol{\Sigma}_P + \boldsymbol{\Sigma}_Q - 2(\boldsymbol{\Sigma}_P^{1/2} \boldsymbol{\Sigma}_Q \boldsymbol{\Sigma}_P^{1/2})^{1/2} \right), \quad (32)$$

which coincides with the Frechét Distance.

Unlike MMD, the Wasserstein distance directly captures the geometry of the underlying space and provides interpretable transport plans between distributions.

## I.3 FRÉCHET DISTANCE FOR GENE EXPRESSION PROFILE EVALUATION

We adapt the Fréchet Inception Distance (FID) framework to evaluate the quality of synthetic gene expression profiles by replacing the Inception network's feature extraction with Principal Component Analysis (PCA). This approach provides a computationally efficient and interpretable metric for comparing distributions of real and synthetic gene expression data.

### I.3.1 PRINCIPAL COMPONENTS CALCULATION

Let $\mathcal{X}_r = \{\mathbf{x}_1^r, \mathbf{x}_2^r, \ldots, \mathbf{x}_n^r\}$ denote the set of real gene expression profiles and $\mathcal{X}_s = (\mathbf{x}_1^s, \mathbf{x}_2^s, \ldots, \mathbf{x}_m^s)$ denote the synthetic profiles, where each $\mathbf{x}_i \in \mathbb{R}^D$ represents the expression levels of $D$ genes.

We first apply PCA to the combined dataset to obtain a projection matrix $\mathbf{W} \in \mathbb{R}^{k \times D}$ containing the top $k$ principal components (e.g., $k = 30$). The feature representations are computed as:

$$\mathbf{z}_i^r = \mathbf{W}^\top \mathbf{x}_i^r, \quad \mathbf{z}_j^s = \mathbf{W}^\top \mathbf{x}_j^s \quad (33)$$

where $\mathbf{z}_i^r, \mathbf{z}_j^s \in \mathbb{R}^k$ are the projected representations in the principal component space.

### I.3.2 FRÉCHET DISTANCE CALCULATION

Assuming the feature representations follow multivariate Gaussian distributions:

- Real data: $\mathcal{N}(\mu_r, \Sigma_r)$;
- Synthetic data: $\mathcal{N}(\mu_s, \Sigma_s)$.

We estimate the parameters:

$$\mu_r = \frac{1}{n} \sum_{i=1}^{n} \mathbf{z}_i^r, \quad \Sigma_r = \frac{1}{n-1} \sum_{i=1}^{n} (\mathbf{z}_i^r - \mu_r)(\mathbf{z}_i^r - \mu_r)^\top \tag{34}$$

$$\mu_s = \frac{1}{m} \sum_{j=1}^{m} \mathbf{z}_j^s, \quad \Sigma_s = \frac{1}{m-1} \sum_{j=1}^{m} (\mathbf{z}_j^s - \mu_s)(\mathbf{z}_j^s - \mu_s)^\top \tag{35}$$

The Fréchet Distance between these distributions is then computed as:

$$\text{FD} = ||\mu_r - \mu_s||_2^2 + \text{Tr}(\Sigma_r + \Sigma_s - 2(\Sigma_r \Sigma_s)^{1/2}) \tag{36}$$

where $\text{Tr}(\cdot)$ denotes the matrix trace and $(\Sigma_r \Sigma_s)^{1/2}$ is the matrix square root of $\Sigma_r \Sigma_s$.

### I.3.3 INTERPRETATION

This metric captures both the difference in means (first term) and the difference in covariance structure (second term) between real and synthetic gene expression profiles in the reduced PCA space. Lower values indicate better agreement between the distributions, suggesting higher quality synthetic data. The use of PCA ensures that the comparison focuses on the most significant sources of variation in the gene expression data while reducing computational complexity from $O(d^2)$ to $O(k^2)$ where typically $k \ll d$.

## I.4 PEARSON CORRELATION COEFFICIENT (PCC)

While MMD and Wasserstein distances measure distributional differences, the Pearson correlation coefficient quantifies linear relationships between paired observations. For two random variables $X$ and $Y$, the population Pearson correlation coefficient is defined as:

$$\rho_{X,Y} = \frac{\text{Cov}(X,Y)}{\sigma_X \sigma_Y} = \frac{\mathbb{E}[(X - \mu_X)(Y - \mu_Y)]}{\sqrt{\mathbb{E}[(X - \mu_X)^2]}\sqrt{\mathbb{E}[(Y - \mu_Y)^2]}}, \tag{37}$$

where $\mu_X, \mu_Y$ are the means and $\sigma_X, \sigma_Y$ are the standard deviations. Given paired samples $\{(x_i, y_i)\}_{i=1}^n$, the sample correlation coefficient is:

$$r = \frac{\sum_{i=1}^{n}(x_i - \bar{x})(y_i - \bar{y})}{\sqrt{\sum_{i=1}^{n}(x_i - \bar{x})^2}\sqrt{\sum_{i=1}^{n}(y_i - \bar{y})^2}}, \tag{38}$$

where $\bar{x} = \frac{1}{n} \sum_{i=1}^{n} x_i$ and $\bar{y} = \frac{1}{n} \sum_{i=1}^{n} y_i$. The coefficient $r \in [-1, 1]$, with $|r| = 1$ indicating perfect linear relationship and $r = 0$ suggesting no linear correlation. For multivariate data $\mathbf{X} \in \mathbb{R}^{n \times d}$ and $\mathbf{Y} \in \mathbb{R}^{n \times d}$, one can compute the average correlation across dimensions or construct a correlation matrix. While Pearson correlation captures only linear dependencies and is sensitive to outliers, it remains widely used due to its computational efficiency and interpretability in assessing feature-wise relationships between datasets.

## I.5 COEFFICIENT OF DETERMINATION ($R^2$ SCORE)

To evaluate the quality of generated single-cell expression profiles, we compute three complementary $R^2$ metrics that capture different statistical properties of the gene expression distributions. Given true counts $\mathbf{X} \in \mathbb{R}^{C \times G}$ and generated counts $\hat{\mathbf{X}} \in \mathbb{R}^{C \times G}$, where $C$ is the number of cells and $G$ is the number of genes, we define:

$R^2$ **Mean** ($R^2_\mu$)    measures how well the model captures the average expression level of each gene across cells:

$$\boldsymbol{\mu}_{\text{true}} = \frac{1}{C} \sum_{c=1}^{C} \mathbf{X}_{c,g}, \quad \boldsymbol{\mu}_{\text{gen}} = \frac{1}{C} \sum_{c=1}^{C} \hat{\mathbf{X}}_{c,g} \tag{39}$$

$$R^2_\mu = 1 - \frac{\sum_{g=1}^{G} (\boldsymbol{\mu}_{\text{true},g} - \boldsymbol{\mu}_{\text{gen},g})^2}{\sum_{g=1}^{G} (\boldsymbol{\mu}_{\text{true},g} - \bar{\boldsymbol{\mu}}_{\text{true}})^2} \tag{40}$$

$R^2$ **Variance** ($R^2_{\sigma^2}$)    measures how well the model captures the variability of expression for each gene:

$$\boldsymbol{\sigma}^2_{\text{true}} = \frac{1}{C} \sum_{c=1}^{C} (\mathbf{X}_{c,g} - \boldsymbol{\mu}_{\text{true},g})^2, \quad \boldsymbol{\sigma}^2_{\text{gen}} = \frac{1}{C} \sum_{c=1}^{C} (\hat{\mathbf{X}}_{c,g} - \boldsymbol{\mu}_{\text{gen},g})^2 \tag{41}$$

$$R^2_{\sigma^2} = 1 - \frac{\sum_{g=1}^{G} (\boldsymbol{\sigma}^2_{\text{true},g} - \boldsymbol{\sigma}^2_{\text{gen},g})^2}{\sum_{g=1}^{G} (\boldsymbol{\sigma}^2_{\text{true},g} - \bar{\boldsymbol{\sigma}}^2_{\text{true}})^2} \tag{42}$$

$R^2$ **Zeros** ($R^2_{\textbf{zeros}}$)    measures how well the model captures the sparsity pattern (dropout rate) for each gene:

$$\mathbf{z}_{\text{true},g} = \sum_{c=1}^{C} \mathbb{1}[\mathbf{X}_{c,g} = 0], \quad \mathbf{z}_{\text{gen},g} = \sum_{c=1}^{C} \mathbb{1}[\hat{\mathbf{X}}_{c,g} = 0] \tag{43}$$

$$R^2_{\text{zeros}} = 1 - \frac{\sum_{g=1}^{G} (\mathbf{z}_{\text{true},g} - \mathbf{z}_{\text{gen},g})^2}{\sum_{g=1}^{G} (\mathbf{z}_{\text{true},g} - \bar{\mathbf{z}}_{\text{true}})^2} \tag{44}$$

These metrics provide a comprehensive assessment of whether the generative model accurately reproduces the first-order statistics (mean), second-order statistics (variance), and sparsity structure (zero-inflation) of the true gene expression distribution at the gene level. Values closer to 1 indicate better agreement between the generated and true distributions. The value is not bounded by -1 but can be arbitrarily ¡ 0.

### I.6    METRICS USED IN EXPERIMENTS

**Reconstruction Metrics**    In our experiments, we use the reconstruction error for the Negative Binomial distribution, PCC and Mean Squared Errors (MSE) as reconstruction metrics.

**Generation Metrics**    For evaluating generation capabilities of models, we use the MMD with the RBF kernel, the Wasserstein Distance, and the Frechet Distance, all calculated to 30 principal components. We compute the PCA on the true data, and project generated data using the loadings. All evaluations were run using 3 generation seeds.

**Perturbation Metrics**    For evaluating perturbation prediction capabilities of models, we used the overlap of DEG (Differentially Expressed Genes) which is related to correlation of average treatment effect ATE-Pearson r in Bereket & Karaletsos (2023), the Spearman Correlation of Log2Fold Changes of significant DEGs, the discrimination L1 score and the Mean Absolute Error (MAE), all computed using the `cell-eval` Adduri et al. (2025) library. All evaluations were run using 3 model outputs, generated for 3 different seeds. For scLDM, we simply sampled from the same model 3 times using different seeds. For all other models, we retrained the model from scratch with different seeds, which only impacts initialization, since the dataset split is fixed.

## J    ABLATION ON TYPE OF CLASSIFIER-FREE GUIDANCE

**Classifier-Free Guidance with Multiple Conditioning Variables.**    In our setting, as described in section 3.2, the diffusion model is conditioned on multiple attributes simultaneously (e.g., cell type and perturbation). We explore two alternative strategies for applying classifier-free guidance (CFG):

**(Type I: Joint conditioning).** A single conditioning token is assigned to each unique combination of attributes. The model output under this strategy is given by

$$\tilde{v}_{t,\epsilon}(\mathbf{Z}, y) = v_{t,\epsilon}(\mathbf{Z}; \text{Null}) + \omega \left[ v_{t,\epsilon}(\mathbf{Z}; y) - v_{t,\epsilon}(\mathbf{Z}; \text{Null}) \right], \tag{45}$$

where $y$ encodes the full joint condition (e.g., "CD4 Naive + IL-9" or "HepG2 + PPP6C").

**(Type II: Additive conditioning).** Instead of encoding combinations directly, we treat each conditioning variable independently. For $M$ attributes with labels $\{y^{(j)}\}_{j=1}^{M}$, the guided output is

$$\tilde{v}_{t,\epsilon}\left(\mathbf{Z}, \{y^{(j)}\}_{j=1}^{M}\right) = v_{t,\epsilon}(\mathbf{Z}; \text{Null}) + \sum_{j=1}^{M} \omega_j \left[ v_{t,\epsilon}(\mathbf{Z}; y^{(j)}) - v_{t,\epsilon}(\mathbf{Z}; \text{Null}) \right], \tag{46}$$

where each attribute contributes an additive adjustment relative to the unconditional prediction.

**Empirical Comparison.** We evaluate both approaches on **Parse 1M** (conditioning on cell type + cytokine perturbation) and **Replogle** (conditioning on cell type + gene knockout). As shown in Table 19, the joint conditioning strategy (Type I) consistently outperforms the additive conditioning strategy (Type II) across metrics, indicating that learning a single joint embedding for each combination of attributes is more effective in capturing complex context–perturbation interactions than treating them independently.

In all experiments, we set the guidance weight $\omega$ to 1, unless specified otherwise, and we did not tune this parameter.

## K    ADDITIONAL RESULTS

### K.1    EXPERIMENT 1: ABLATION EXPERIMENTS AND ADDITIONAL RESULTS

In Table 14 we report an ablation study on encoding gene expression following our approach of zero padding the non-expressed genes (see Appendix E.1), or utilizing the full context as input, as typically done in scVI. Our approach is superior in terms of reconstruction performance, as well as more computationally efficient.

Table 14: Reconstruction performance comparison of our scLDM using the zero padding strategy for encoding, or using all genes as input.

| Dataset | Model | RE $\downarrow$ | PCC $\uparrow$ | MSE $\downarrow$ |
|---------|-------|------|------|------|
| Dentate Gyrus | full context | 5458.6 | 0.097 | 0.252 |
| | zero padding | **5325.3** | **0.125** | **0.242** |

In Figure 4, we report another ablation study on hyperparameters of the VAE architecture: we sweep over depth (number of layers for the encoder/decoder layers), width (embedding dimensions) and number of latent tokens, over two datasets that consists of vastly different number of genes (18k for dentate gyrus versus 2k for replogle) and cells (600k for replogle and 18k for dentate gyrus). Interestingly, depth appears to negatively impact performance in the dentate gyrus dataset, whereas it does not in the Replogle dataset. For both datasets, a larger embedding dimension as well as a higher number of latent tokens improve performance.

In Figure 5, we compare the performance of our latent aggregation approach against set-based aggregation methods (max, mean, and sum pooling) across two datasets (dentate gyrus in the top row and Replogle in the bottom row).

We implemented the Set-Transformer-based aggregation in the following way: Given an input tensor $\mathbf{X} \in \mathbb{R}^{B \times G \times D}$, where $B$ is the batch size, $G$ is the number of genes, and $D$ is the embedding dimension, the module computes:

$$\mathbf{Z} = \text{MLP}(\text{Agg}(\mathbf{X})^{\top}) \tag{47}$$

where the aggregation function $\text{Agg} : \mathbb{R}^{B \times G \times D} \to \mathbb{R}^{B \times D \times 1}$ is defined as:

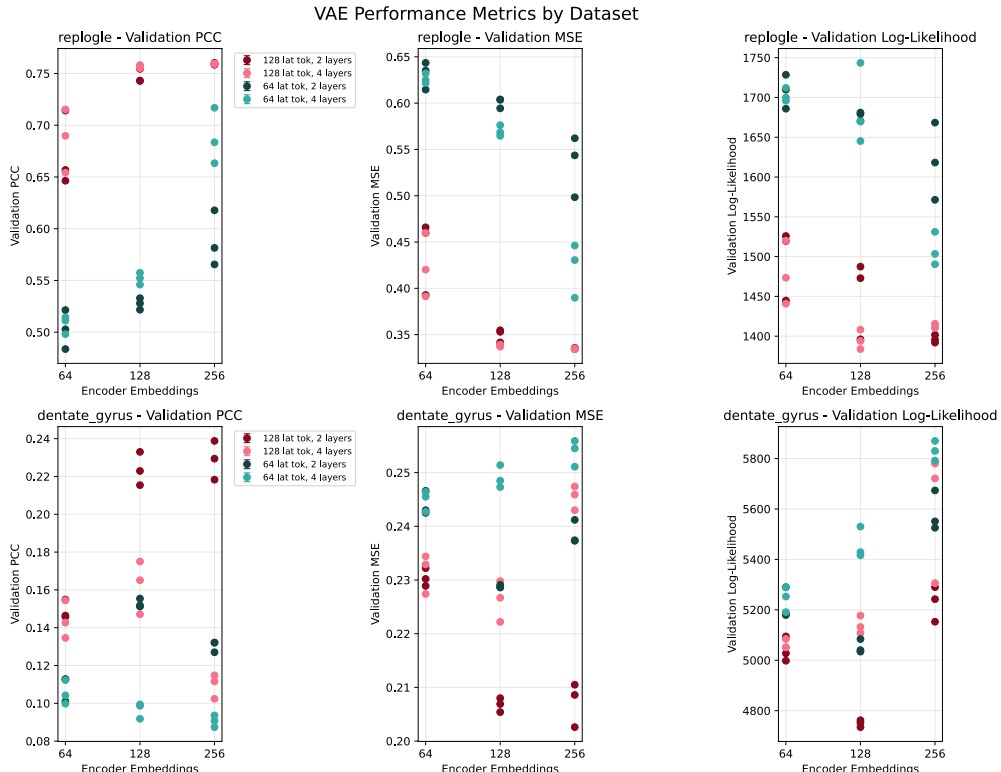

Figure 4: Ablations on VAE width, depth and number of latent tokens

$$\text{Agg}(\mathbf{X})_{b,d,1} = \begin{cases} \max_{g=1}^{G} \mathbf{X}_{b,g,d} & \text{if aggregation\_type = ``max"} \\ \frac{1}{G} \sum_{g=1}^{G} \mathbf{X}_{b,g,d} & \text{if aggregation\_type = ``mean"} \\ \sum_{g=1}^{G} \mathbf{X}_{b,g,d} & \text{if aggregation\_type = ``sum"} \end{cases} \tag{48}$$

After transposition, the MLP consists of a linear projection followed by layer normalization:

$$\text{MLP}(\mathbf{Y}) = \text{LayerNorm}(\mathbf{YW} + \mathbf{b}) \tag{49}$$

where $\mathbf{W} \in \mathbb{R}^{1 \times D'}$ and $\mathbf{b} \in \mathbb{R}^{D'}$ are learnable parameters with $D' = \texttt{n\_embed}$. The final output $\mathbf{Z} \in \mathbb{R}^{B \times D \times D'}$.

This ensures that the dimensionality of the latent space, as well as the number of parameters for the encoder-decoder transformer layers is comparable between this implementation and the latent implementation.

We evaluate each aggregation method using three reconstruction metrics: log-likelihood, mean squared error (MSE), and Pearson correlation coefficient (PCC). Results are shown for two model configurations with embedding dimensions of 64 and 128, with error bars representing the standard deviation across three random seeds. The high errors bars for the mean and sum aggregation in the replogle datasets are due to high training instability that we observed in the training runs, making some runs to temporarily diverge.

In Table 15, we present the model comparison on benchmark datasets, but on all genes. Please note that in this comparison, scDiffusion performs better than in the table on the metrics computed only on highly variable genes. The reason for that is that the data is extremely sparse, and the model synthesizes data consisting mostly of zeros. As a result, the match becomes better. This is a clear indication of deficiencies of the currently used evaluation metrics for generative models, which is a long-standing issue in the field (Theis et al., 2016).

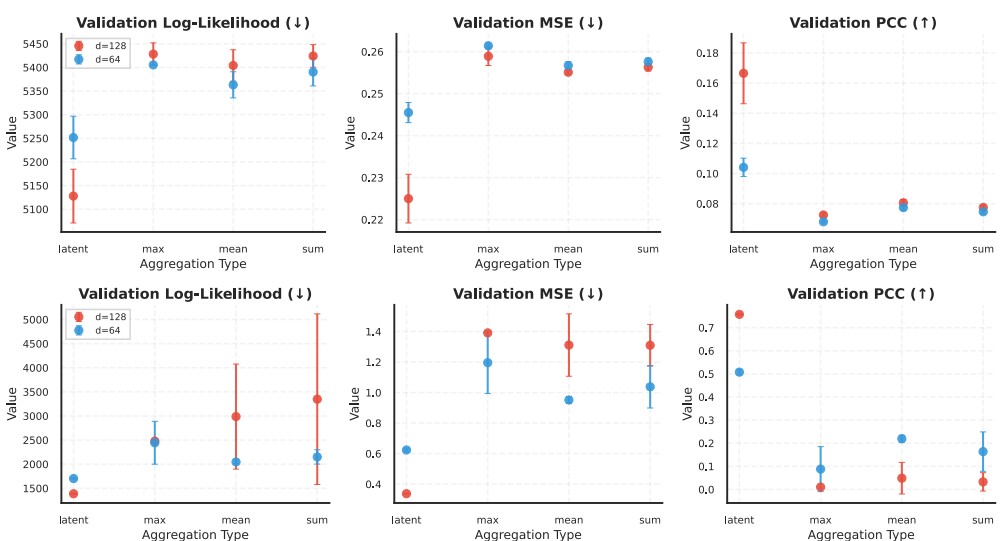

Figure 5: Ablations on different aggregation strategies between Set-Transformer-like max, sum and mean aggregation, versus the perceiver-style latent aggregation used in our model. Top: dentate gyrus dataset, Bottom: replogle dataset.

Table 15: Model performance comparison on unconditional and conditional cell generation benchmarks on all genes. W2, MMD$^2$ RBF and FD metrics calculated on 30 principal components are reported (lower is better).

| Setting | Model | W2 ↓ | MMD$^2$ RBF ↓ | FD ↓ |
|---------|-------|------|---------------|------|
| | | Dentate Gyrus | | |
| | scDiffusion | $8.545 \pm 0.061$ | $0.021 \pm 0.000$ | $16.303 \pm 0.265$ |
| Uncond | CFGen | $11.396 \pm 0.025$ | $0.027 \pm 0.001$ | $24.942 \pm 0.456$ |
| | Ours | $9.489 \pm 0.054$ | $0.027 \pm 0.000$ | $28.307 \pm 0.357$ |
| | scDiffusion | $8.906 \pm 0.041$ | $0.093 \pm 0.001$ | $28.829 \pm 1.612$ |
| Cond | CFGen | $10.580 \pm 0.022$ | $0.082 \pm 0.001$ | $40.298 \pm 0.472$ |
| | Ours | $9.147 \pm 0.024$ | $0.108 \pm 0.004$ | $31.045 \pm 0.884$ |
| | | Tabula Muris | | |
| | scDiffusion | $8.616 \pm 0.215$ | $0.002 \pm 0.000$ | $6.881 \pm 0.565$ |
| Uncond | CFGen | $11.331 \pm 0.081$ | $0.009 \pm 0.000$ | $31.788 \pm 1.073$ |
| | Ours | $10.573 \pm 0.092$ | $0.005 \pm 0.000$ | $17.641 \pm 0.337$ |
| | scDiffusion | $11.459 \pm 0.081$ | $0.035 \pm 0.001$ | $43.456 \pm 1.678$ |
| Cond | CFGen | $9.420 \pm 0.041$ | $0.026 \pm 0.001$ | $22.045 \pm 0.389$ |
| | Ours | $8.530 \pm 0.110$ | $0.019 \pm 0.001$ | $15.547 \pm 0.557$ |
| | | HLCA | | |
| | scDiffusion | $9.234 \pm 0.008$ | $0.002 \pm 0.000$ | $5.585 \pm 0.180$ |
| Uncond | CFGen | $12.651 \pm 0.025$ | $0.008 \pm 0.000$ | $24.038 \pm 0.492$ |
| | Ours | $10.816 \pm 0.089$ | $0.010 \pm 0.000$ | $24.126 \pm 0.473$ |
| | scDiffusion | $9.998 \pm 0.048$ | $0.094 \pm 0.002$ | $40.093 \pm 3.103$ |
| Cond | CFGen | $10.715 \pm 0.039$ | $0.087 \pm 0.005$ | $36.178 \pm 0.961$ |
| | Ours | $9.350 \pm 0.046$ | $0.084 \pm 0.005$ | $28.398 \pm 1.358$ |

In Table 16 we present results on $R^2$ scores for all genes in the observational datasets, for unconditional and conditional settings.

Table 16: Model performance on $R^2$ metrics across datasets for unconditional and conditional generation.

| Dataset | Setting | Model | $R^2$ **Mean** ↑ | $R^2$ **Var** ↑ | $R^2$ **Zeros** ↑ |
|---|---|---|---|---|---|
| Dentate Gyrus | Unconditional | cfgen | $1.00 \pm 0.00$ | $0.99 \pm 0.00$ | $1.00 \pm 0.00$ |
| | | scDiffusion | $1.00 \pm 0.00$ | $-1.17 \pm 0.02$ | $< -10$ |
| | | scLDM | $0.99 \pm 0.00$ | $0.96 \pm 0.00$ | $0.96 \pm 0.00$ |
| | Conditional | cfgen | $1.00 \pm 0.00$ | $0.99 \pm 0.00$ | $1.00 \pm 0.00$ |
| | | scDiffusion | $1.00 \pm 0.00$ | $-1.14 \pm 0.02$ | $< -10$ |
| | | scLDM | $0.99 \pm 0.00$ | $0.96 \pm 0.00$ | $0.96 \pm 0.00$ |
| HLCA | Unconditional | cfgen | $1.00 \pm 0.00$ | $0.99 \pm 0.00$ | $0.99 \pm 0.00$ |
| | | scDiffusion | $1.00 \pm 0.00$ | $0.10 \pm 0.00$ | $< -10$ |
| | | scLDM | $0.99 \pm 0.00$ | $0.97 \pm 0.00$ | $0.87 \pm 0.00$ |
| | Conditional | cfgen | $1.00 \pm 0.00$ | $0.99 \pm 0.00$ | $1.00 \pm 0.00$ |
| | | scDiffusion | $1.00 \pm 0.00$ | $0.07 \pm 0.01$ | $< -10$ |
| | | scLDM | $0.99 \pm 0.00$ | $0.97 \pm 0.00$ | $0.88 \pm 0.00$ |
| Tabula Muris | Unconditional | cfgen | $1.00 \pm 0.00$ | $0.98 \pm 0.00$ | $0.99 \pm 0.00$ |
| | | scDiffusion | $1.00 \pm 0.00$ | $0.53 \pm 0.00$ | $< -10$ |
| | | scLDM | $0.99 \pm 0.00$ | $0.97 \pm 0.00$ | $0.95 \pm 0.00$ |
| | Conditional | cfgen | $1.00 \pm 0.00$ | $0.99 \pm 0.00$ | $0.99 \pm 0.00$ |
| | | scDiffusion | $1.00 \pm 0.00$ | $0.56 \pm 0.01$ | $< -10$ |
| | | scLDM | $0.99 \pm 0.00$ | $0.97 \pm 0.00$ | $0.95 \pm 0.00$ |

In Figure 6, 7, and 8, a visualization of gene-wise variance for true and generated data on Dentate Gyrus, Tabula Muris, and HLCA, respectively, in the conditional settings. CFGen and scLDM properly recover true variance, with a slight tendency of scLDM to overestimate, while scDiffusion completely fails and underestimates the true Variance.

In Figure 9, we present a visualization of true and generated data for all models and all datasets in UMAP coordinates. In Figures 10, 11, 12, we present the conditional generation results in UMAP coordinates colored by the conditional class.

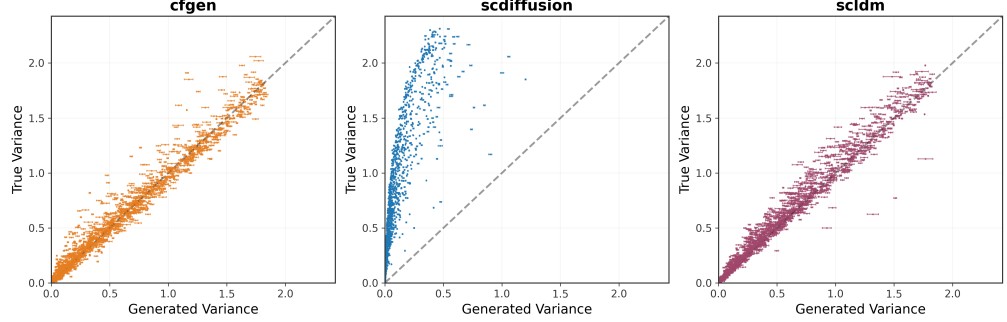

Figure 6: Visualization of the gene-wise variance for true and generated data for CFGen (left), scDiffusion (middle) our model (right), for the conditional generation settings on Dentate Gyrus. The error bars represent the standard errors over 3 seeds.

## K.2 EXPERIMENT 1: INTERPRETABILITY RESULTS ON CROSS-ATTENTION SCORES

The cross-attention encoder and decoder layers can be interpreted as pooling and unpooling operators on the gene tokens at input and output. The attention scores of the cross attention layers provide an interpretability tool to analyze how gene tokens map to the latent tokens. We analyzed the attention patterns of the encoder and decoder cross-attention layers in the dentate gyrus dataset. To this end, we computed the marker genes for each cell type annotated in the dataset. We further extracted the attention weights between the gene tokens and the latent tokens and averaged them across 1k

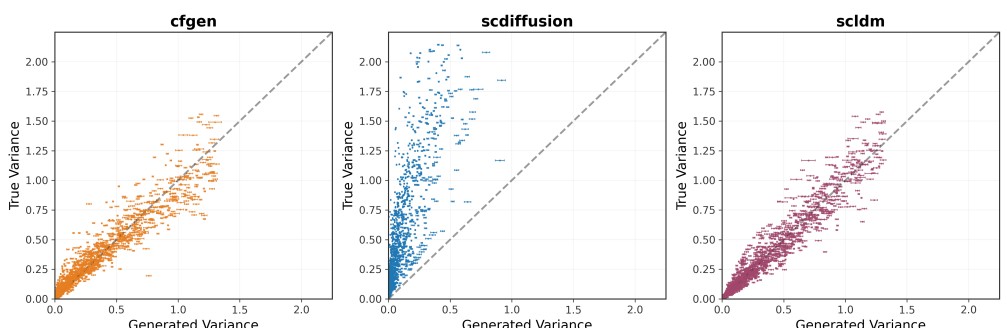

Figure 7: Visualization of the gene-wise variance for true and generated data for CFGen (left), scDiffusion (middle) our model (right), for the conditional generation settings on Tabula Muris. The error bars represent the standard errors over 3 seeds.

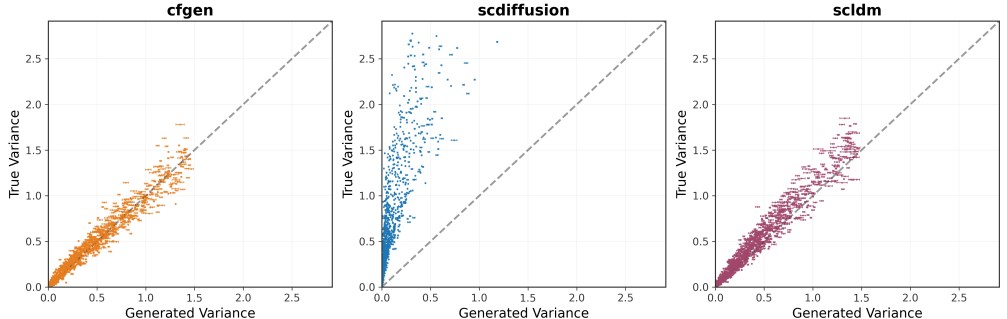

Figure 8: Visualization of the gene-wise variance for true and generated data for CFGen (left), scDiffusion (middle) our model (right), for the conditional generation settings on HLCA. The error bars represent the standard errors over 3 seeds.

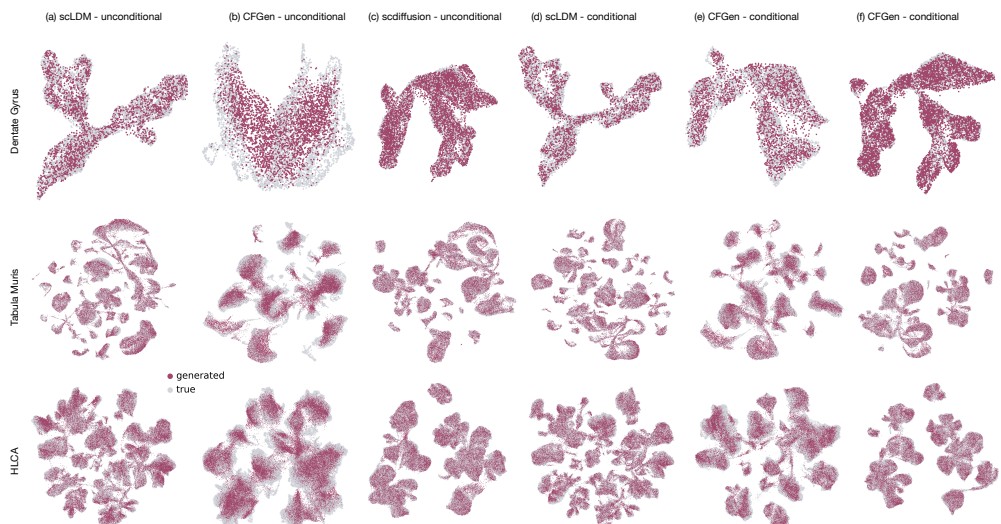

Figure 9: Visualization of the generation results for all datasets and models for conditional and unconditional generations. True and Generated gene expression is embedded in UMAP coordinates jointly, upon normalization, following standard Scanpy pipeline.

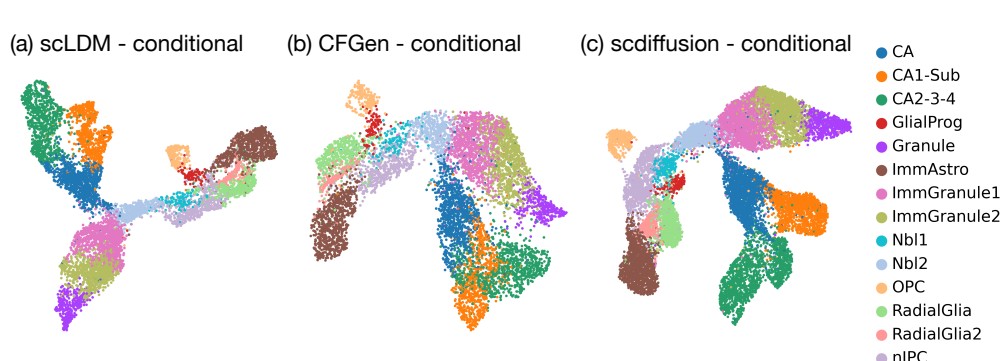

Figure 10: Visualization of the conditional generation results for the dentate gyrus dataset and all models, colored by the conditional label (clusters).

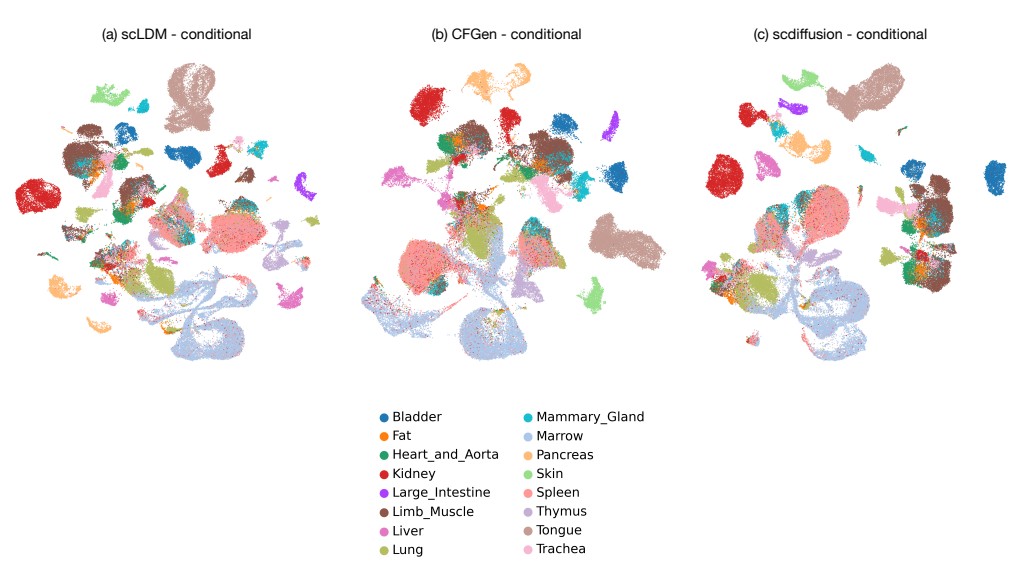

Figure 11: Visualization of the conditional generation results for the tabula muris dataset and all models, colored by the conditional label (tissue).

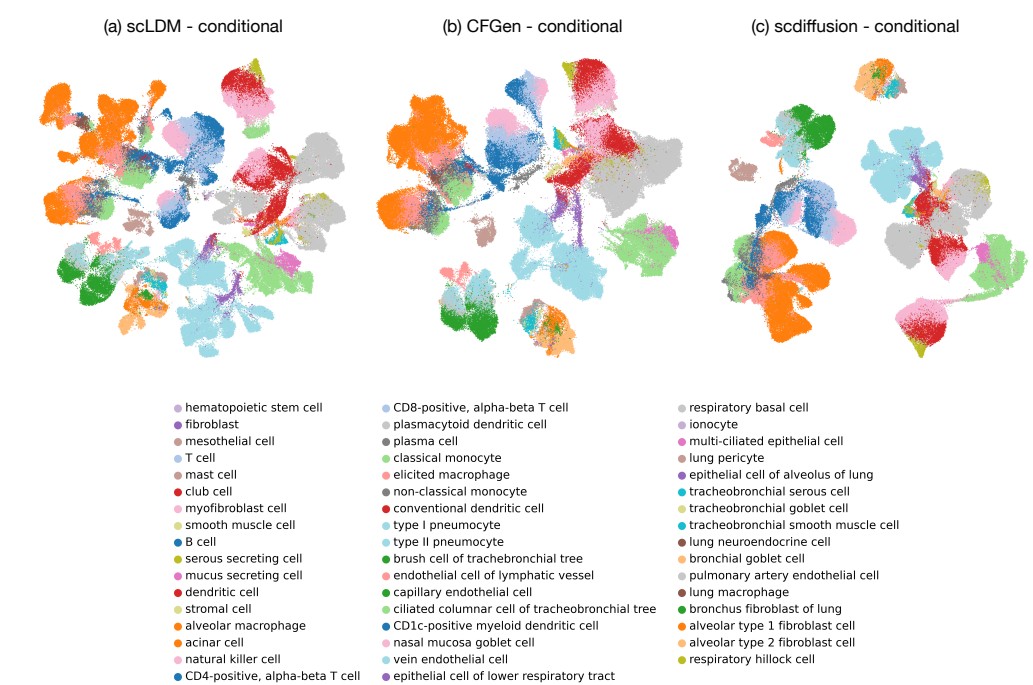

Figure 12: Visualization of the conditional generation results for the hlca dataset and all models, colored by the conditional label (cell type).

cells (where each celltype was sampled roughly the same proportion with respect to the original dataset). Using the genes v. latent tokens average attention matrix, we computed enrichment scores for the cell type markers for each latent token using decoupler Badia-I-Mompel et al. (2022). We visualized the enrichment score between each latent token and cell-type marker genes in Figure 13, where each column (latent tokens) and rows (marker genes gene set) was clustered using hierarchical clustering. We can observe that the latent tokens do show a selective enrichment for marker genes of specific cell types. In particular, the enrichment score seems to separate correctly the groups of cell types belonging to the Neuron, Oligodendrocyte and Glia lineage, both in the encoder and decoder layer.

### K.3 EXPERIMENT 2: RECONSTRUCTION CAPABILITIES ON PERTURBATION DATASETS

The results presented in Table 17 demonstrate that our proposed approach significantly outperforms the scVI baseline across all evaluated metrics on the Parse1M and Replogle dataset. Most notably, our method achieves a substantially lower reconstruction error (RE) of about $310$ nats compared to scVI's $432$ nats, indicating better reconstructive capabilities. Furthermore, our approach yields a remarkable improvement in Pearson correlation coefficient (PCC), achieving $0.887$ versus scVI's mediocre $0.351$, which suggests that our model captures the underlying biological relationships much more effectively. The mean squared error (MSE) is also greatly reduced from $0.701$ to $0.188$, representing an approximately $73\%$ reduction in reconstruction error. These consistent improvements across multiple evaluation criteria provide strong evidence that our method offers substantial advantages over scVI and indicate its great potential in analyzing biological data.

We further report gene-level $R^2$ scores for all perturbation datasets and models in Table 18. Our model shows competitivbe performance for the $R^2$ mean dramatic improved performance for the $R^2$ variance in the perturbation conditional generation task.

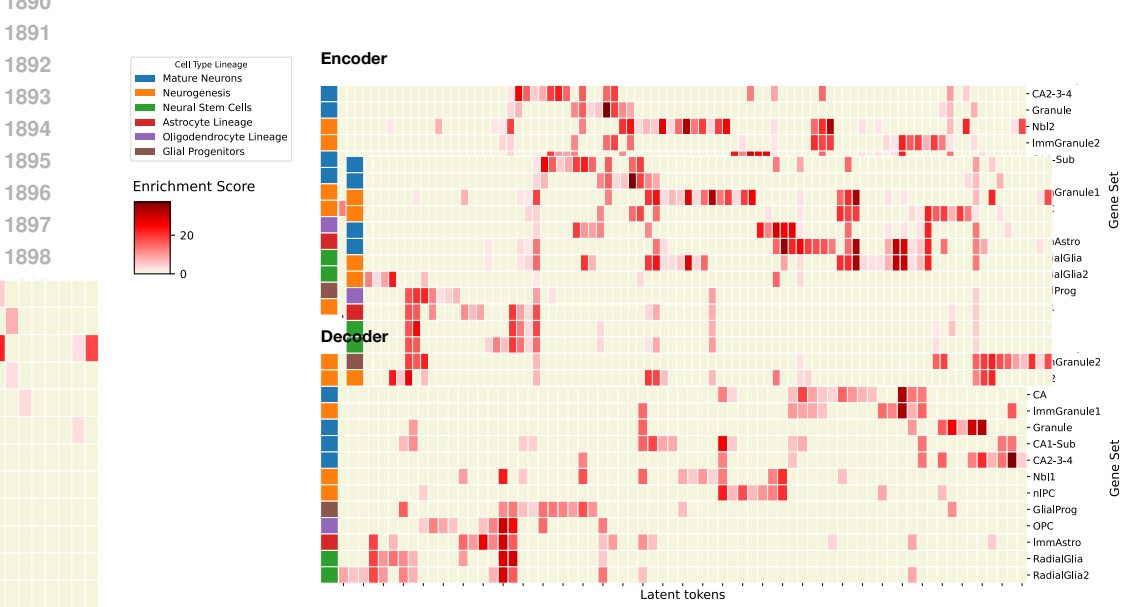

Figure 13: Enerichment scores for marker genes of cell-types in the dataset of cross-attention for both the cross-attention encoder and decoder layers.

Table 17: Model performance comparison on cell reconstruction task.

| Dataset | Model | RE ↓ | PCC ↑ | MSE ↓ |
|---|---|---|---|---|
| Parse 1M | scVI | $432.41 \pm 0.08$ | $0.351 \pm 0.000$ | $0.701 \pm 0.001$ |
| | scLDM | $\mathbf{149.70 \pm 0.22}$ | $\mathbf{0.874 \pm 0.003}$ | $\mathbf{0.165 \pm 0.002}$ |
| Replogle | scVI | $2144.86 \pm 0.35$ | $0.166 \pm 0.000$ | $0.703 \pm 0.001$ |
| | scLDM | $\mathbf{1590.51 \pm 0.38}$ | $\mathbf{0.713 \pm 0.004}$ | $\mathbf{0.285 \pm 0.002}$ |

Table 18: Model performance on $R^2$ metrics across datasets.

| Dataset | Model | $R^2$ Mean ↑ | $R^2$ Var ↑ | $R^2$ Zeros ↑ |
|---|---|---|---|---|
| Parse 1M | Cellflow | $1.00 \pm 0.00$ | $0.39 \pm 0.00$ | $< -10$ |
| | CPA | $1.00 \pm 0.00$ | $< -10$ | $< -10$ |
| | scGPT | $-1.17 \pm 0.02$ | $< -10$ | $< -10$ |
| | scLDM ($\omega$=1) | $0.96 \pm 0.00$ | $0.98 \pm 0.00$ | $0.92 \pm 0.00$ |
| | scLDM ($\omega$=5) | $0.96 \pm 0.00$ | $0.96 \pm 0.00$ | $0.94 \pm 0.00$ |
| | scLDM ($\omega$=10) | $0.94 \pm 0.00$ | $0.94 \pm 0.00$ | $0.91 \pm 0.00$ |
| | scVI | $-0.27 \pm 0.06$ | $< -10$ | $< -10$ |
| | STATE | $-3.40 \pm 0.09$ | $< -10$ | $0.90 \pm 0.00$ |
| Replogle | Cellflow | $0.99 \pm 0.00$ | $< -10$ | $0.00 \pm 0.00$ |
| | CPA | $1.00 \pm 0.00$ | $< -10$ | $0.00 \pm 0.00$ |
| | scGPT | $< -10$ | $< -10$ | $0.00 \pm 0.00$ |
| | scLDM ($\omega$=1) | $0.99 \pm 0.00$ | $0.87 \pm 0.00$ | $0.87 \pm 0.00$ |
| | scLDM ($\omega$=5) | $0.98 \pm 0.00$ | $0.86 \pm 0.00$ | $0.87 \pm 0.00$ |
| | scLDM ($\omega$=10) | $0.97 \pm 0.00$ | $0.78 \pm 0.00$ | $0.86 \pm 0.00$ |
| | scVI | $0.91 \pm 0.00$ | $< -10$ | $0.00 \pm 0.00$ |
| | STATE | $0.51 \pm 0.02$ | $-8.81 \pm 4.35$ | $0.80 \pm 0.05$ |

### K.4 EXPERIMENT 2: A COMPARISON BETWEEN *additive* AND *joint* CONDITIONING IN CLASSIFIER-FREE GUIDANCE

Table 19 compares the performance of our scLDM model using two different classifier-free guidance approaches for conditional cell generation: the additive steering method proposed by Palma et al. (2025a) and our joint attribute steering method. Across all metrics (Wasserstein-2 distance, $MMD^2$ RBF, and Fréchet Distance) and both datasets (Parse 1M and Replogle), the joint approach consistently outperforms the additive approach, demonstrating substantial improvements in generation quality.

Table 19: Model performance comparison on conditional cell generation on Parse1M and Replogle. For these results scLDM was trained using the classifier-free guidance approach proposed in Palma et al. (2025a) (additive) and ours (joint).

| Dataset | Model | W2 $\downarrow$ | $MMD^2$ RBF $\downarrow$ | FD $\downarrow$ |
|---|---|---|---|---|
| Parse 1M | scLDM (additive) | $15.850 \pm 0.073$ | $0.129 \pm 0.004$ | $109.196 \pm 2.933$ |
| | scLDM (joint) | $\mathbf{12.455} \pm 0.001$ | $\mathbf{0.027} \pm 0.000$ | $\mathbf{18.145} \pm 0.068$ |
| Replogle | scLDM (additive) | $18.538 \pm 0.058$ | $0.451 \pm 0.003$ | $255.510 \pm 2.163$ |
| | scLDM (joint) | $\mathbf{11.288} \pm 0.011$ | $\mathbf{0.200} \pm 0.001$ | $\mathbf{53.555} \pm 0.210$ |

### K.5 EXPERIMENT 2: PERTURBATION PREDICTION METRICS

**Perturbation results using pertrubation metrics from the `cell-eval` package.** We further evaluated our models and all baselines on the generated results for the test set perturbations using the `cell-eval` package [7] (Figure 14). Our model, although not explicitly designed for perturbation prediction, is competitive across various metrics compared to the baselines on both datasets considered.

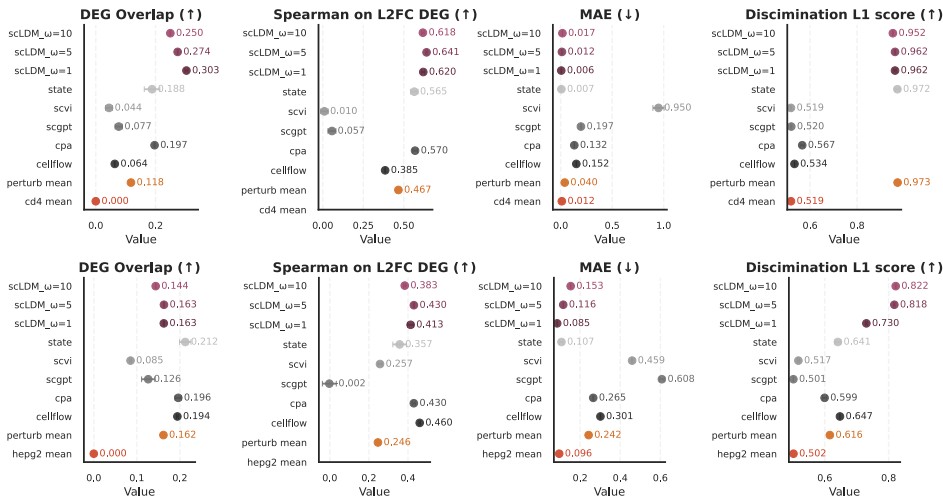

Figure 14: Evaluation metrics from the `cell-eval` package for the Parse 1M dataset (top) and Replogle dataset (bottom)

**Generation metrics on Differentially Expressed Genes** Table 20 presents generation metrics for the same datasets and models as in Table 3, but evaluated specifically on Differentially Expressed Genes (DEGs). To identify DEGs, we applied Scanpy's `tl.rank_genes_groups` method using the Wilcoxon rank-sum test for each perturbation. For perturbations with more than 10 identified DEGs, we computed generation metrics directly in the gene space rather than in PCA-reduced space. This evaluation offers insight into the model's performance on the most biologically relevant genes affected by each perturbation.

---

[7] https://github.com/ArcInstitute/cell-eval

Table 20: Generation metrics computed on differentially expressed gene space for the same datasets and models shown in main results.

| Dataset | Model | W2 ↓ | MMD$^2$ RBF ↓ | FD ↓ |
|---------|-------|------|---------------|------|
| Parse 1M | scVI | $26.131 \pm 0.721$ | $1.112 \pm 0.013$ | $703.413 \pm 39.084$ |
| | CPA | $\underline{18.168} \pm 0.426$ | $1.378 \pm 0.015$ | $342.468 \pm 16.009$ |
| | Cellflow | $\mathbf{17.990} \pm 0.516$ | $0.036 \pm 0.002$ | $114.441 \pm 10.376$ |
| | scGPT | $25.853 \pm 0.513$ | $1.813 \pm 0.025$ | $687.085 \pm 27.790$ |
| | STATE | $21.304 \pm 0.506$ | $0.576 \pm 0.005$ | $346.857 \pm 16.616$ |
| | scLDM ($\omega$=1) | $20.155 \pm 0.620$ | $\mathbf{0.008} \pm 0.001$ | $\mathbf{21.255} \pm 1.733$ |
| | scLDM ($\omega$=5) | $20.015 \pm 0.633$ | $\underline{0.025} \pm 0.002$ | $\underline{62.641} \pm 4.938$ |
| | scLDM ($\omega$=10) | $20.434 \pm 0.650$ | $0.050 \pm 0.004$ | $106.257 \pm 7.866$ |
| Replogle | scVI | $12.587 \pm 0.180$ | $0.440 \pm 0.003$ | $184.379 \pm 4.090$ |
| | CPA | $\underline{9.871} \pm 0.131$ | $0.756 \pm 0.006$ | $114.196 \pm 2.715$ |
| | Cellflow | $\mathbf{9.311} \pm 0.131$ | $0.338 \pm 0.004$ | $\mathbf{81.365} \pm 2.468$ |
| | scGPT | $27.690 \pm 0.342$ | $3.043 \pm 0.011$ | $892.139 \pm 20.073$ |
| | STATE | $15.390 \pm 0.203$ | $0.549 \pm 0.006$ | $236.127 \pm 5.047$ |
| | scLDM ($\omega$=1) | $11.428 \pm 0.179$ | $\mathbf{0.176} \pm 0.002$ | $\underline{97.500} \pm 3.825$ |
| | scLDM ($\omega$=5) | $11.727 \pm 0.179$ | $\underline{0.218} \pm 0.003$ | $111.085 \pm 3.701$ |
| | scLDM ($\omega$=10) | $12.656 \pm 0.191$ | $0.276 \pm 0.004$ | $140.190 \pm 4.274$ |

## K.6 EXPERIMENT 3

For the last experiment, we trained three VAEs for our approach (scLDM-VAE): with 20M parameters, 70M parameters, and 270M parameters. Further, we evaluated the resulting models using embeddings on a downstream task (classification) for two out-of-distribution datasets (COVID-19 and Tabula Sapiens 2.0).

First, we evaluated these three versions of our model using reconstruction metrics on the dataset they were trained on, namely, Human Census Data from CellxGene[8]. Looking at Table 21, we can see a clear relationship between model size and reconstruction performance for the scLDM-VAE models on the CellxGene dataset. As the number of parameters increases from 20M to 270M, all three metrics show substantial improvement: reconstruction error (RE) decreases, Pearson correlation coefficient (PCC) increases, and mean squared error (MSE) drops. These results demonstrate that scaling up the scLDM-VAE architecture yields consistent performance gains across all reconstruction metrics, with the 270M parameter model achieving approximately 17% lower reconstruction error and 18% higher correlation compared to the smallest 20M model.

Table 21: Reconstruction performance comparison of our scLDM-VAEs with varying number of parameters: 20M, 70M, and 270M.

| Dataset | Model | RE ↓ | PCC ↑ | MSE ↓ |
|---------|-------|------|-------|-------|
| CellxGene Census | scLDM-VAE (20M) | 1742.7 | 0.661 | 0.137 |
| | scLDM-VAE (70M) | 1552.7 | 0.739 | 0.106 |
| | scLDM-VAE (270M) | **1441.7** | **0.783** | **0.091** |

Table 22 presents a comprehensive performance comparison of various models on the COVID-19 dataset, averaged across all donors. Our scLDM model with 270M parameters achieves the best performance across all metrics (ROC AUC, PR AUC, F1 Score, Recall, and Precision), demonstrating consistent improvements over both transformer-based baselines (TranscriptFormer, scGPT, Geneformer, UCE) and traditional VAE approaches (scVI, AIDO.Cell).

Figure 15 visualizes the receiver operating characteristic (ROC) and precision-recall (PR) curves for all models on the COVID-19 classification task. The curves further illustrate the superior discriminative performance of scLDM variants, with the 270M parameter model achieving the highest area under both curves, consistent with the quantitative results in Table 22.

---

[8] https://cellxgene.cziscience.com/

Table 22: COVID-19 Model Performance Summary (Averaged Across All Donors). **Bold** indicates the best performing model.

| Model | ROC AUC | PR AUC | F1 Score | Recall | Precision |
|---|---|---|---|---|---|
| scLDM (270M) | **0.909**± 6e-04 | **0.877**± 0.001 | **0.820**± 0.001 | **0.836**± 0.001 | **0.806**± 0.001 |
| TranscriptFormer | 0.905± 4e-04 | 0.874± 9e-04 | 0.814± 0.002 | 0.829± 0.003 | 0.801± 0.001 |
| scLDM (70M) | 0.905± 5e-04 | 0.872 ± 0.001 | 0.815 ± 0.001 | 0.83 ± 0.002 | 0.801 ± 0.001 |
| scLDM (20M) | 0.902± 5e-04 | 0.869 ± 0.001 | 0.811 ± 0.001 | 0.827 ± 0.001 | 0.797 ± 0.002 |
| UCE | 0.876± 5e-04 | 0.834± 0.002 | 0.775± 8e-04 | 0.781± 0.001 | 0.771± 0.002 |
| scGPT | 0.876± 4e-04 | 0.831± 0.001 | 0.779± 9e-04 | 0.793± 0.002 | 0.766± 0.001 |
| Geneformer | 0.866± 6e-04 | 0.815± 0.001 | 0.768± 0.001 | 0.781± 0.003 | 0.757± 0.001 |
| AIDO.Cell | 0.821± 7e-04 | 0.753± 9e-04 | 0.717± 8e-04 | 0.729± 0.002 | 0.708± 0.001 |
| scVI | 0.800± 7e-04 | 0.709± 0.001 | 0.675± 0.001 | 0.680± 0.002 | 0.680± 0.001 |

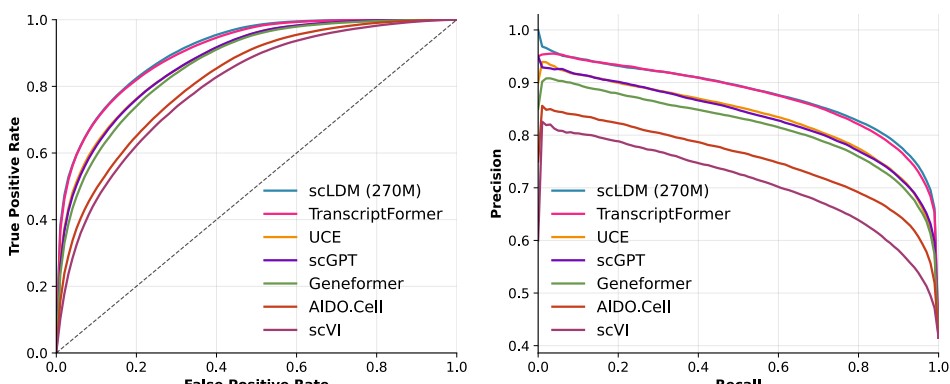

Figure 15: Precision-recall and receiver operator curves for COVID-19 data.

Table 23 summarizes model performance on the Tabula Sapiens 2.0 dataset, averaged across all tissues. Notably, the smallest scLDM variant (20M parameters) achieves the highest F1 score (0.804), slightly outperforming both larger scLDM models and all baseline methods, suggesting that model scale may have diminishing returns on this particular cell type classification task.

Table 23: Tabula Sapiens 2.0 model performance summary (averaged across all tissues)

| Model | F1 Score | Recall | Precision |
|---|---|---|---|
| scLDM-20M | **0.804** ± 0.002 | **0.805** ± 0.002 | **0.812** ± 0.002 |
| scLDM-270M | 0.802 ± 0.002 | 0.803 ± 0.002 | 0.811 ± 0.002 |
| scLDM-70M | 0.802 ± 0.002 | 0.802 ± 0.002 | 0.810 ± 0.002 |
| scGPT | 0.800 ± 0.002 | 0.802 ± 0.002 | 0.806 ± 0.002 |
| scVI | 0.799 ± 0.002 | 0.794 ± 0.002 | 0.814 ± 0.003 |
| TranscriptFormer | 0.799 ± 0.002 | 0.800 ± 0.002 | 0.802 ± 0.002 |
| UCE | 0.796 ± 0.002 | 0.797 ± 0.001 | 0.801 ± 0.003 |
| Geneformer | 0.777 ± 0.002 | 0.776 ± 0.002 | 0.786 ± 0.003 |
| AIDO.Cell | 0.724 ± 0.002 | 0.715 ± 0.002 | 0.748 ± 0.003 |