# OpenReview forum: "Scalable Single-Cell Gene Expression Generation with Latent Diffusion Models"
_ICLR.cc/2026/Conference — Submitted to ICLR 2026_

### Official Review · Reviewer_rHzp · 2025-10-18

**Soundness:** 2
**Presentation:** 3
**Contribution:** 2
**Rating:** 4
**Confidence:** 3

**Summary:**

The paper proposes scLDM, a two‑stage generative framework for single‑cell gene expression that (i) uses a fully transformer VAE designed for exchangeable sets of genes and (ii) replaces the VAE’s Gaussian prior with a latent diffusion model trained with flow matching and linear interpolants (SiT) and parameterized with Diffusion Transformers (DiT). The key architectural element is a Multi‑head Cross‑Attention Block (MCAB) that’s used for permutation‑invariant pooling in the encoder and permutation‑equivariant “unpooling” in the decoder via index‑selected embeddings. The decoder parametrizes a Negative Binomial likelihood for counts. Experiments cover: (1) reconstruction on observational scRNA‑seq, (2) unconditional and conditional generation on observational data, (3) multi‑attribute conditional generation on perturbational datasets (cell context × perturbation), and (4) downstream classification using the learned embeddings. In tables and plots, scLDM typically outperforms scVI, CFGen, CPA, and scDiffusion on W2/MMD²/Fréchet and reconstruction metrics; the classifier‑free guidance is extended to multi‑hot conditioning. Figure 1 gives the system diagram; Tables 1–3 report core wins; Figure 2–3 show qualitative UMAPs; Table 4 summarizes downstream classification.

**Strengths:**

- Exchangeability by construction. The encoder’s MCAB uses fixed “pseudo‑inputs” $S$ to pool arbitrary gene sets into a fixed number of latent tokens; the decoder restores per‑gene outputs by cross‑attending from latents to the index‑selected embeddings $E_I$. This yields permutation‑invariant posteriors and permutation‑equivariant likelihoods, matching the set‑valued nature of genes. The design is clean and scalable relative to naïve all‑token attention.
- Multi‑attribute conditioning at sampling time. The extended classifier‑free guidance supports joint multi‑hot attributes, not just additive singletons, and the ablation in the supplement claims the standard joint form outperforms the additive variant for multi‑attribute control.
- Downstream utility. VAE embeddings competitive with scRNA representation models on COVID‑19 classification; F1 up to 0.820 with the largest scLDM encoder, on par with TranscriptFormer and better than scVI/scGPT/Geneformer.

**Weaknesses:**

- Novelty is incremental on the set‑architecture axis. MCAB is essentially a Perceiver‑style cross‑attention with learned inducing tokens in the encoder and index‑selected keys in the decoder. The “unified block for pooling/unpooling” is neat, but the paper underplays overlap with SetTransformer/Perceiver‑family parameterizations. The exchangeability story is correct but not conceptually new; the new part is applying this cleanly to scRNA with a counts‑likelihood, then pairing it with a DiT latent prior. The framing should be more honest about what’s new vs re‑packaged.
- Evaluation uses HVGs in several core tables, partially undercutting the “scalable to all genes” narrative. For perturbational datasets, analyses restrict to the top 2,000 HVGs. This weakens the claim that scLDM intrinsically handles arbitrarily large, unordered gene sets without feature selection.
- Metrics and calibration risk. W2, MMD²‑RBF, and Fréchet on embeddings are common, but the paper doesn’t show per‑gene marginal calibration (e.g., KS or QQ vs Negative Binomial), zero‑inflation fidelity, or gene‑gene correlation structure retention. The qualitative UMAPs are persuasive but can be misleading without per‑marker distribution checks.

**Questions:**

- (Related work) "two key challenges limit existing methods." The author ignore some recent work like scGPT, scFoundation, scdiffusion, etc, that are not based on GAN. The claim of the second limit is thus invalid. Additionally, related statistical generative models/simulators like Splatter are not compared.
- Even though the gene order in general datasets does not has specific biological meaningful, their corresponding location on DNA could provide some information, if one orders the genes in this way.
- What are the computational complexities for compared methods? Provide FLOPs/throughput and memory scaling with number of genes, latent tokens m, and pseudo‑input size $∣S∣$, for both encoder/decoder and the DiT prior. Include wall‑clock on representative datasets.
- Exchangeability baselines. Please run an ablation comparing MCAB to SetTransformer/Perceiver variants with and without the index‑selected $E_I$ trick. Are the gains architectural or due to the latent diffusion prior? Report reconstruction and generation deltas.

---

> ### Author Response · Authors · 2025-11-25
> **Rebuttal (1/2)**
>
> # General
>
> We would like to thank the reviewer for their time spent on our paper and for providing their comments. Please find our responses below.
>
> # Responses
>
> **Re “Novelty is incremental on the set‑architecture axis. The framing should be more honest about what’s new vs re‑packaged”.**
>
> Thank you for your comment. Assessing the novelty of a paper is always a controversial topic at the AI conferences (https://medium.com/@black_51980/novelty-in-science-8f1fd1a0a143). We disagree with calling our approach incremental. For instance, Appendix E.1 outlines how our model deals with *dropouts*, enforcing the latent space to implicitly incorporate the information about non-expressed genes. However, we do agree that some statements should be downplayed a bit. We accommodated the reviewer’s suggestion in the text by being more precise about the novelty (e.g., removed “novelty” about leveraging MCAB in line 169).
>
> **Re “Evaluation uses HVGs in several core tables, partially undercutting the “scalable to all genes” narrative. For perturbational datasets, analyses are restricted to the top 2,000 HVGs. This weakens the claim that scLDM intrinsically handles arbitrarily large, unordered gene sets without feature selection.”**
>
> Thank you for this remark. Our model can handle arbitrarily large, unordered gene sets without feature selection. In fact, for the first experiment on observational datasets, the VAE transformer architecture proposed by us is able to scale to ~30k genes in context, both at the input and output of the autoencoder. The way the evaluation is carried out is still an open question in the field, and we do not want to dispute about this issue in our paper. We follow typical procedures and comply with them. Similarly, we do not fully agree with using 30 principal components in calculating some of the metrics, but this is how other papers assessed their models (some accepted at ICLR). Therefore, we respectfully disagree with the reviewer in this regard, and we do not agree that the evaluation procedure in any way influences any of the properties of our approach.
>
> **Re “Metrics and calibration risk. W2, MMD²‑RBF, and Fréchet on embeddings are common, but the paper doesn’t show per‑gene marginal calibration (e.g., KS or QQ vs Negative Binomial), zero‑inflation fidelity, or gene‑gene correlation structure retention. The qualitative UMAPs are persuasive but can be misleading without per‑marker distribution checks.”**
>
> We thank the reviewer for the comment. We have added $R^2$ score (coefficient of determination) for generated cells on all datasets and models tested. Please look at Table 16 and 18 in the appendix, as well as Appendix H, for a description of how they are computed. These are $R^2$ scores for the mean, variance, and zeroes across all genes, for each cell. They are effectively a way to measure the consistency of the marginal distribution between true and generated data. As can be seen, our model is on par with other models in terms of $R^2$ score for the means across genes, but it’s superior to baselines for $R^2$ score for the gene variance as well as zeros, which is a measure of a faithfully the sparsity is recovered.
>
>
> **Re “(Related work) "two key challenges limit existing methods." The authors ignore some recent work like scGPT, scFoundation, scdiffusion, etc, that are not based on GAN. The claim of the second limit is thus invalid. Additionally, related statistical generative models/simulators like Splatter are not compared.”**
>
> We thank the reviewer for the comment. We actually do compared to scdiffusion (see Table 2 in the main text, as well as several additional results and metrics in Appendix K) for observational data, and we introduced scGPT (both chemical and genetic) as a new baseline (see Table 3 in the main text, as well as additional results in Appendix K). We did not compare to scGAN since it was already reported by Palma et al. that it was not as strong a baseline as their model, and we did not find a GAN-based approach for a perturbation prediction-type model. We would like to highlight that our model is superior to both baselines (scdiffusion and scgpt), across all metrics and datasets that we compared against.
> We also do not plan to compare against synthetic data, using tools like Splatter, as we have decided to leave this for future work.

---

> > ### Author Response · Authors · 2025-11-25
> > **Rebuttal (2/2)**
> >
> > **Re “Even though the gene order in general datasets does not has specific biological meaningful, their corresponding location on DNA could provide some information, if one orders the genes in this way.”**
> >
> > We thank the reviewer for the comment. We believe it does not provide much biological inductive bias, since Gene Regulatory Network architecture is determined by very complex genomic arrangements, drives chromatin contacts and transcription factors that give rise to 3D structures of the chromosomes. See for instance Dekker et al. 2016 https://www.cell.com/cell/fulltext/S0092-8674(16)30073-3
> > We leave it to future work to integrate epigenomics data in the design of a single cell generative model architecture.
> >
> > **Re “What are the computational complexities for compared methods? Provide FLOPs/throughput and memory scaling with number of genes, latent tokens m, and pseudo‑input size , for both encoder/decoder and the DiT prior. Include wall‑clock on representative datasets.”**
> >
> > Thank you for your comment. Below, we provide a table with the required information:
> >
> > | Dataset | VAE FLOPs | LDM FLOPs | VAE Params | LDM Params |
> > |---------|-----------|-----------|------------|------------|
> > | dentate_gyrus | 5.02 GFLOPs | 6.45 GFLOPs | 3.45M | 9.77M |
> > | tabula_muris | 29.45 GFLOPs | 77.33 GFLOPs | 13.23M | 57.81M |
> > | hlca | 38.13 GFLOPs | 77.33 GFLOPs | 15.35M | 57.83M |
> > | replogle | 28.02 GFLOPs | 51.56 GFLOPs | 21.16M | 39.91M |
> > | parse1m | 28.02 GFLOPs | 51.56 GFLOPs | 21.16M | 38.93M |
> >
> > Similarly, for COVID-19 and Tabula Sapiens 2.0:
> >
> > | Model | VAE FLOPs | VAE Params |
> > |-------|-----------|------------|
> > | VAE_Census_20M | 0.33 TFLOPs | 23.75M |
> > | VAE_Census_70M | 1.02 TFLOPs | 76.07M |
> > | VAE_Census_270M | 2.63 TFLOPs | 270.14M |
> >
> > We added this to Appendix H.
> >
> > **Re “Exchangeability baselines. Please run an ablation comparing MCAB to SetTransformer/Perceiver variants with and without the index‑selected trick. Are the gains architectural or due to the latent diffusion prior? Report reconstruction and generation deltas.”**
> >
> > Thank you for this suggestion. We ran additional experiments on Dentate Gyrus and Replogle and compared our MCAB to three other aggregation operators, namely, max, mean and sum. We provide the results on that in Figure 5 (Appendix K.1).

---

### Official Review · Reviewer_1AGg · 2025-10-29

**Soundness:** 2
**Presentation:** 2
**Contribution:** 2
**Rating:** 4
**Confidence:** 3

**Summary:**

The paper proposes scLDM, a fully transformer-based VAE for exchangeable single-cell gene expression that uses a Multi-head Cross-Attention Block (MCAB) both as a permutation-invariant pooling operator in the encoder and a permutation-equivariant unpooling operator in the decoder. The standard Gaussian prior is replaced with a latent diffusion model (DiT + SiT with flow matching) to improve sample quality and enable multi-attribute classifier-free guidance. Empirically, scLDM is evaluated on observational datasets (Dentate Gyrus, Tabula Muris, HLCA) and perturbational datasets (Parse 1M, Replogle), reporting strong reconstruction and generation results and competitive downstream classification from VAE embeddings.

**Strengths:**

* The paper is well-written and well-motivated.
* They showed competitive or superior performance of their method across different experiments.
* The idea of MCAB in the single-cell generative model context is novel.

**Weaknesses:**

* The text, especially in the method section and experiment description, lacks clarity, includes incorrect cross-referencing, and overloaded notation (e.g, equation 33). The trains and test sets across experiments are not well specified.
 * In Table 1, the HLCA Pearson correlation (~0.1 for other methods vs. ~0.4 for scLDM) does not appear consistent with the UMAPs in Figure 2.
* Metrics and baselines for the perturbation benchmark are limited; more recent methods (e.g., CellFlow, GEARS) should be included. Because perturbation effects are subtle, it is important to report metrics on differentially expressed genes (DEGs).
* Results in Tables 1 and 2 do not match the CFGen paper on the same benchmarks; this may be due to hyperparameters or data splits and should be reconciled.

**Questions:**

* The method comprises several components, but there is no ablation in the main text. What is the effect of each component (e.g., MCAB design, latent diffusion prior, guidance strategy)?
* Were the foundation models in the benchmark fine-tuned on the training data, or used zero-shot? Please clarify.

---

> ### Author Response · Authors · 2025-11-25
> **Rebuttal**
>
> # General
>
> We would like to thank the reviewer for their time spent on our paper and for providing their comments. Please find our responses below.
>
> # Responses
>
> **Re “The text lacks clarity and some notations are overloaded.”**
>
> Thank you for your comment. We went through the text and did our best to correct some descriptions. Please check the Appendix for further details, e.g., regarding how train and test sets are formulated. For the observational data, we used exactly the same train/test splits as Palma et al. (CFGen), for the perturbation data we believe are already adequately described, and follow the approach from Adduri et al. (STATE), whereas for the CXG Census experiment we clarified in the Appendix K.6
>
> **Re “In Table 1, the HLCA Pearson correlation (~0.1 for other methods vs. ~0.4 for scLDM) does not appear consistent with the UMAPs in Figure 2.”**
>
> We would like to clarify that PCC is used for **reconstruction** purposes, i.e., it measures how well a variational autoencoder reconstructs real data, while Figure 2 indicates conditional generations. As a result, it is not possible to compare those two results. Nevertheless, we believe that the UMAP qualitative report provides a much better coverage of the generated data for our models, as it faithfully captures the density of smaller sub-clusters. In contrast, CFGen and scDiffusion only cover the main clusters, losing the sub-cluster structure.
>
> **Re “Metrics and baselines for the perturbation benchmark are limited; more recent methods.”**
>
> Thank you for your comment. We fully agree with the reviewer. Therefore, we provided three new methods: STATE, scGPT and CellFlow (see Table 3). Moreover, we added some metrics from cell-eval, namely, DEG Overlap, Spearman on L2FC DEG, MAE, Discrimination L1 score, in Figure 14 (Appendix K). Our model, although not specifically designed for perturbation prediction, demonstrates on-par or superior performance across the four metrics and two datasets compared to the baselines.
>
> **Re “Results in Tables 1 and 2 do not match the CFGen paper on the same benchmarks; this may be due to hyperparameters or data splits and should be reconciled.”**
>
> Thank you for spotting that. This is true and we are also puzzled by these numbers. We used checkpoints provided in the repository of CFGen to be as accurate as possible and to avoid any unnecessary modeling misalignment. We had an internal debate about approaching the authors of the paper and further inquiring about this peculiarity, but we decided that this would potentially violate the double-blind review process. However, we assume the reason lies in the manner data was transformed and how PCA was calculated. Nevertheless, we conducted our evaluations in a consistent manner for all models to ensure that the entire evaluation is fair to all models. We would also like to point out that we included scDiffusion both for conditional and unconditional settings, as opposed to the CFGen comparison.
>
> **Re “The method comprises several components, but there is no ablation in the main text. What is the effect of each component (e.g., MCAB design, latent diffusion prior, guidance strategy)?”**
>
> Thank you for this suggestion. We ran additional experiments on Dentate Gyrus and Replogle and compared our MCAB to three other aggregation operators, namely, max, mean and sum. We provide the results on that in Figure 5 (Appendix K.1).
>
>
> **Re “Were the foundation models in the benchmark fine-tuned on the training data, or used zero-shot? Please clarify.”**
>
> None of the foundation models were trained on the two datasets presented; hence, this constitutes an OOD evaluation.

---

### Official Review · Reviewer_7TKA · 2025-11-02

**Soundness:** 2
**Presentation:** 2
**Contribution:** 2
**Rating:** 2
**Confidence:** 2

**Summary:**

The paper introduces a two-stage generative model for single-cell RNA-seq. First, a transformer VAE with a single Multi-head Cross-Attention Block (MCAB) encodes unordered gene sets and decodes them back, staying permutation-aware. Second, instead of a Gaussian prior, the latent space is modeled with a Diffusion Transformer trained with flow matching and classifier-free guidance, so the model can generate cells under multiple conditions (cell type, context, perturbation).

**Strengths:**

1. Doing the diffusion in the latent tokens learned by the VAE avoids operating over tens of thousands of genes. Using DiT in latent space is interesting.

**Weaknesses:**

1. The paper says: train VAE -> freeze -> train DiT diffusion with classifier-free guidance. But we don’t get wall-clock, number of diffusion steps, model sizes per dataset, or memory footprint, yet the model is transformer-based end to end and is evaluated on million-cell datasets. This matters for reproducibility and to support the “scalable” claim.
2. The MCAB is the key novelty, but we don’t see a direct ablation like “replace MCAB with SetTransformer pooling” or “use separate pooling/unpooling” to isolate how much of the gain comes from this block vs. the diffusion prior.
3. Comparison set is strong but not exhaustive. scLDM is compared to CPA and scVI on perturbations, but there are more recent OOD-perturbation models (e.g. transport/OT-style, flow-matching-on-manifolds).
4. Biological faithfulness is shown mostly through distributional metrics + UMAPs. W2, MMD², FD are good, but we need lineage/trajectory or marker-level checks (for example “does the model preserve marker co-expression under perturbation?”).

**Questions:**

1. If you replace MCAB with (i) pooling by multihead attention (SetTransformer-style) or (ii) a Perceiver IO block with separate latent arrays, how much do reconstruction and FD/W2 degrade?
2. You report better F1 than scGPT/Geneformer on COVID. Is this purely because the VAE is trained on a big Human Census dataset, or is there a benefit from diffusion pretraining too?

---

> ### Author Response · Authors · 2025-11-25
> **Rebuttal**
>
> # General
>
> We would like to thank the reviewer for their time spent on our paper and for providing their comments. Please find our responses below.
>
> # Responses
>
> **Re “No wall-clock, diffusion steps, models sizes per dataset, memory footprint, etc.”.**
>
> Thank you for your comment. Below, we provide a table with the required information:
>
> | Dataset | VAE FLOPs | LDM FLOPs | VAE Params | LDM Params |
> |---------|-----------|-----------|------------|------------|
> | dentate_gyrus | 5.02 GFLOPs | 6.45 GFLOPs | 3.45M | 9.77M |
> | tabula_muris | 29.45 GFLOPs | 77.33 GFLOPs | 13.23M | 57.81M |
> | hlca | 38.13 GFLOPs | 77.33 GFLOPs | 15.35M | 57.83M |
> | replogle | 28.02 GFLOPs | 51.56 GFLOPs | 21.16M | 39.91M |
> | parse1m | 28.02 GFLOPs | 51.56 GFLOPs | 21.16M | 38.93M |
>
> Similarly, for COVID-19 and Tabula Sapiens 2.0:
>
> | Model | VAE FLOPs | VAE Params |
> |-------|-----------|------------|
> | VAE_Census_20M | 0.33 TFLOPs | 23.75M |
> | VAE_Census_70M | 1.02 TFLOPs | 76.07M |
> | VAE_Census_270M | 2.63 TFLOPs | 270.14M |
>
> We added this to Appendix H.
>
> **Re “No ablation for MCAB”**
>
> Thank you for this suggestion. We ran additional experiments on Dentate Gyrus and Replogle dataset and compared our MCAB to three other aggregation operators, namely, max, mean and sum. We provide the results on that in Figure 5 (Appendix K.1).
>
> **Re “More comparisons beyond CPA”.**
>
> Thank you for your comment. We highly appreciate your comment and agree with it. Since comparing to some models is non-trivial without a public code repository, we managed to add a couple of new methods, namely, STATE, scGPT, and CellFlow. Please see Table 3 in Section 4.2, as well as Appendix K.5 with new metrics for the perturbation baselines.
>
> **Re “Biological faithfulness is shown mostly through distributional metrics”**
>
> We appreciate this comment because this is indeed important to present biological faithfulness . For this purpose, we presented visualizations of the gene-wise variance for true and generated data, UMAPs of conditionally generated cell in Appendix K.1. Moreover, now we added new results on cell-eval perturbation metrics for the perturbation datasets in Appendix K.5, and visualized some interpretability results in the latent cross-attention patterns in Appendix K.2.  We also report generation results only on DEG genes in Table 20.
>
> **Re “You report better F1 than scGPT/Geneformer on COVID. Is this purely because the VAE is trained on a big Human Census dataset, or is there a benefit from diffusion pretraining too?”**
>
> We want to highlight that for COVID-19, we trained only the VAE part, without the diffusion. Therefore, we claim that the advantage comes from two facts: (i) our new architecture, (ii) and the ability to scale our architecture to the Human Census data.

---

### Author Response · Authors · 2025-11-25
**General response (1/2)**

## General
We thank all reviewers for their constructive and thoughtful feedback. We appreciate the time and effort invested in evaluating our submission and for raising points that have helped us improve the clarity, completeness, and impact of our work. In response to the comments, we have substantially expanded the experiments, clarified methodological choices, added detailed computational and biological analyses, and refined the presentation of our contributions.

## Summary of Changes
To address reviewer concerns, we made several key updates and additions to the paper:

1. *Comprehensive Computational Analysis*: We added detailed FLOP counts, parameter sizes, and comparisons across all datasets and models (Appendix H), as well as memory and scalability analyses.
2. *Expanded Ablation and Baseline Studies*:  We introduced new ablation experiments for the Multi-Head Cross-Attention block (MCAB) and compared it against other aggregation mechanisms (max, mean, sum, see Figure 5, Appendix K.1), as well as additional baselines such as STATE, scGPT, and CellFlow (See Appendix K.5 as well as Table 3 in main text).
3. *Enhanced Biological Faithfulness and Interpretability*: We included new visualizations of gene-wise variance (Table 16 and 18 in Appendix K), perturbation evaluation metrics (cell-eval metrics, Figure 14 in Appendix K.5), and latent cross-attention interpretability analyses (Figure 13, Appendix K.2).
4. *Clarified Experimental Setup and Evaluation Protocols*: We specified that all foundation models were evaluated zero-shot, provided additional context on training/test splits, and ensured fair evaluation across all baselines.
5. *Improved Framing*:  We adjusted framing to better communicate the novelty and scope of our contributions.

We believe these updates significantly strengthen the paper’s technical depth and clarity.

## Responses to Reviewers

### Computational Complexity and Model Scaling

We thank the reviewers for their request for computational comparisons. Below, we summarize FLOPs and parameter counts for all datasets:

| Dataset | VAE FLOPs | LDM FLOPs | VAE Params | LDM Params |
|---------|-----------|-----------|------------|------------|
| dentate_gyrus | 5.02 GFLOPs | 6.45 GFLOPs | 3.45M | 9.77M |
| tabula_muris | 29.45 GFLOPs | 77.33 GFLOPs | 13.23M | 57.81M |
| hlca | 38.13 GFLOPs | 77.33 GFLOPs | 15.35M | 57.83M |
| replogle | 28.02 GFLOPs | 51.56 GFLOPs | 21.16M | 39.91M |
| parse1m | 28.02 GFLOPs | 51.56 GFLOPs | 21.16M | 38.93M |

Similarly, for larger-scale models:
| Model | VAE FLOPs | VAE Params |
|-------|-----------|------------|
| VAE_Census_20M | 0.33 TFLOPs | 23.75M |
| VAE_Census_70M | 1.02 TFLOPs | 76.07M |
| VAE_Census_270M | 2.63 TFLOPs | 270.14M |

These results are now reported in Appendix H. The added analysis highlights that our method scales gracefully across datasets and model sizes.

### Ablations and Exchangeability Baselines

To better understand the contributions of our MCAB design, we ran controlled experiments on Dentate Gyrus and Replogle comparing MCAB with alternative set aggregation operators (max, mean, sum). Results are reported in Figure 5 (Appendix K.1).

These experiments demonstrate that MCAB provides consistent improvements, validating its architectural contribution beyond the diffusion prior.

We also included additional ablation assessing how hyperparameters such as number of latent tokens, embedding dimension, and number of layers, impact reconstruction performance. Results can be found in Figure 4 (Appendix K.1).

### Expanded Baselines and Perturbation Metrics

In response to requests for broader comparisons, we incorporated STATE, scGPT, and CellFlow as new baselines (see Table 3, Section 4.2).

We also added new cell-eval perturbation metrics, such as DEG Overlap, Spearman on L2FC DEG, MAE, and Discrimination L1 score, shown in Figure 14 (Appendix K). We also report generation metrics results on DEGs in Table 20 (Appendix K.5). Our model matches or outperforms the baselines across four metrics and two datasets, despite not being specifically designed for perturbation prediction.

### Biological Faithfulness and Interpretability

We expanded our biological validation through:
- Gene-wise mean, variance, and sparsity comparisons between real and generated data (Appendix K.2).
- Latent cross-attention interpretability visualizations (Appendix K.2).
- Perturbation evaluation metrics (Appendix K.5).

These additions confirm that the model captures biologically meaningful structure, not merely distributional similarity.

### COVID-19 & Tabula Sapiens 2.0 Evaluation

For the COVID-19 and Tabula Sapiens 2.0 datasets, we trained only the VAE component without diffusion pretraining. The superior F1 performance is attributed to:

- The novel architectural design, and
- Its scalability to large-scale Human Census data.

This clarifies that gains do not originate from diffusion training.

---

> ### Author Response · Authors · 2025-11-25
> **General response (2/2)**
>
> ### Evaluation Protocol and Model Novelty
>
> All foundation models were evaluated zero-shot, i.e., without any fine-tuning on the benchmark datasets, constituting an out-of-distribution (OOD) test scenario.
>
> We acknowledge the reviewer’s comment on the novelty framing and have moderated our claims accordingly. Nonetheless, our contribution is not incremental:
>
> - The model introduces mechanisms (Appendix E.1) for encoding gene dropouts into the latent space, a feature absent in prior works.
> - The combination of exchangeable set modeling with latent diffusion priors remains novel.
>
> We have revised the manuscript to convey this balance clearly and transparently.
>
> ### Scalability and Use of HVGs
>
> Our architecture inherently supports arbitrarily large, unordered gene sets. As reported Table 6, Appendix I, our model scales to up to 27,997 genes (in the case of HLCA dataset), using the same set of genes as used by previous baselines. In the evaluation for the observational datasets (section 4.1 of the manuscript), we report metrics only for HVG because they more faithfully evaluate the generated gene expression profiles. Nevertheless, we also report the same generation metrics for all genes in Table 15, Appendix K.2, where our model is still competitive against the baselines, and for large datasets, it outperforms the baselines. However, due to the sparsity of the data, we believe that reporting the metrics in HVG space is more meaningful since it captures real biological variation.
>
> For the perturbation dataset, we adhere to community standards, namely, using the top 2,000 HVGs, to ensure comparability with prior work. These evaluation choices do not affect the fundamental scalability or expressiveness of our model, which is still superior in several (but not all) instances. Yet, we believe this is a less informative evaluation due to high sparsity (see discussion in Appendix K.1).
>
>
> ## Closing Remarks
>
> We again thank all reviewers for their insightful feedback. In summary, our revisions include:
>
> - A comprehensive computational and architectural analysis (Appendix H).
> - New ablations and baselines (Tables 3, Figures 4,5 & 14, Table 20).
> - Expanded biological faithfulness and interpretability results (Appendix K, Table 16,18, Figure 13).
>
> We believe these extensive updates address all major reviewer concerns and significantly strengthen the paper’s technical and empirical contributions.

---

### Meta-Review · Area_Chair_1PmQ · 2025-12-05

**Summary:**

1. Scalability evidence: reviewers wanted FLOPs, parameter counts, memory and wall-clock data – provided in rebuttal.

2. Architectural novelty: MCAB was seen as incremental; rebuttal added MCAB-vs-baseline pooling ablation but not vs SetTransformer/Perceiver.

3. Evaluation scope: main tables use 2 k HVGs, weakening “arbitrary-gene-set” claim; full-gene results relegated to appendix.

4. Biological fidelity: requests for gene-wise marginals/zero-inflation met with R² metrics, yet gene-gene covariance/trajectory checks still missing.

5. Missing baselines: STATE, scGPT, CellFlow and cell-eval metrics added; diffusion-step timing and full-gene main-table remain outstanding.

**Reviewer Concerns:**

Addressed: FLOPs/params, MCAB ablation vs max/mean/sum, new baselines (STATE/scGPT/CellFlow), gene-wise R², zero-shot clarification, data-split details.

Still out: wall-clock/diffusion-step numbers, full-gene (non-HVG) tables in main text, direct MCAB vs SetTransformer/Perceiver ablation, gene-gene covariance checks.

**Reviewer Scores:**

The reviewers are unlikely to raise their scores significantly.

---

### Decision · Program_Chairs · 2026-01-26

Reject